# An Exact Characterization of the Generalization Error for the Gibbs Algorithm

**Gholamali Aminian** *
University College London
g.aminian @ucl.ac.uk

**Yuheng Bu** *
Massachusetts Institute of Technology
buyuheng@mit.edu

**Laura Toni**
University College London
l.toni@ucl.ac.uk

**Miguel Rodrigues**
University College London
m.rodrigues@ucl.ac.uk

**Gregory Wornell**
Massachusetts Institute of Technology
gww@mit.edu

## Abstract

Various approaches have been developed to upper bound the generalization error of a supervised learning algorithm. However, existing bounds are often loose and lack of guarantees. As a result, they may fail to characterize the exact generalization ability of a learning algorithm. Our main contribution is an exact characterization of the expected generalization error of the well-known Gibbs algorithm (a.k.a. Gibbs posterior) using symmetrized KL information between the input training samples and the output hypothesis. Our result can be applied to tighten existing expected generalization error and PAC-Bayesian bounds. Our approach is versatile, as it also characterizes the generalization error of the Gibbs algorithm with data-dependent regularizer and that of the Gibbs algorithm in the asymptotic regime, where it converges to the empirical risk minimization algorithm. Of particular relevance, our results highlight the role the symmetrized KL information plays in controlling the generalization error of the Gibbs algorithm.

## 1  Introduction

Evaluating the generalization error of a learning algorithm is one of the most important challenges in statistical learning theory. Various approaches have been developed [55], including VC dimension-based bounds [66], algorithmic stability-based bounds [16], algorithmic robustness-based bounds [72], PAC-Bayesian bounds [45], and information-theoretic bounds [71].

However, upper bounds on generalization error may not entirely capture the generalization ability of a learning algorithm. One apparent reason is the tightness issue, some upper bounds [9] can be far away from the true generalization error or even vacuous when evaluated in practice. More importantly, existing upper bounds do not fully characterize all the aspects that could influence the generalization error of a supervised learning problem. For example, VC dimension-based bounds depend only on the hypothesis class, and algorithmic stability-based bounds only exploit the properties of the learning algorithm. As a consequence, both methods fail to capture the fact that generalization error depends strongly on the interplay between the hypothesis class, learning algorithm, and the

---

*Equal contribution

35th Conference on Neural Information Processing Systems (NeurIPS 2021).

underlying data-generating distribution, as discussed in [71, 73]. This paper overcomes the above limitations by deriving an exact characterization of the generalization error for a specific supervised learning algorithm, namely the Gibbs algorithm.

## 1.1 Problem Formulation

Let $S = \{Z_i\}_{i=1}^n$ be the training set, where each $Z_i$ is defined on the same alphabet $\mathcal{Z}$. Note that $Z_i$ is not required to be i.i.d generated from the same data-generating distribution $P_Z$, and we denote the joint distribution of all the training samples as $P_S$. We denote the hypotheses by $w \in \mathcal{W}$, where $\mathcal{W}$ is a hypothesis class. The performance of the hypothesis is measured by a non-negative loss function $\ell : \mathcal{W} \times \mathcal{Z} \to \mathbb{R}_0^+$, and we can define the empirical risk and the population risk associated with a given hypothesis $w$ as

$$L_E(w, s) \triangleq \frac{1}{n} \sum_{i=1}^n \ell(w, z_i), \quad \text{and} \quad L_P(w, P_S) \triangleq \mathbb{E}_{P_S}[L_E(w, S)], \tag{1}$$

respectively. A learning algorithm can be modeled as a randomized mapping from the training set $S$ onto an hypothesis $W \in \mathcal{W}$ according to the conditional distribution $P_{W|S}$. Thus, the expected generalization error quantifying the degree of over-fitting can be written as

$$\overline{\text{gen}}(P_{W|S}, P_S) \triangleq \mathbb{E}_{P_{W,S}}[L_P(W, P_S) - L_E(W, S)], \tag{2}$$

where the expectation is taken over the joint distribution $P_{W,S} = P_{W|S} \otimes P_S$.

In this paper we focus on the generalization error of the Gibbs algorithm (a.k.a. Gibbs posterior [21]). The $(\alpha, \pi(w), f(w, s))$-Gibbs distribution, which was first proposed by [28] in statistical mechanics and further investigated by [35] in information theory, is defined as:

$$P_{W|S}^\alpha(w|s) \triangleq \frac{\pi(w) e^{-\alpha f(w,s)}}{V(s, \alpha)}, \quad \alpha \geq 0, \tag{3}$$

where $\alpha$ is the inverse temperature, $\pi(w)$ is an arbitrary prior distribution of $W$, $f(w, s)$ is energy function, and $V(s, \alpha) \triangleq \int \pi(w) e^{-\alpha f(w,s)} dw$ is the partition function.

If $P$ and $Q$ are probability measures over space $\mathcal{X}$, and $P$ is absolutely continuous with respect to $Q$, the Kullback-Leibler (KL) divergence between $P$ and $Q$ is given by $D(P\|Q) \triangleq \int_{\mathcal{X}} \log\left(\frac{dP}{dQ}\right) dP$. If $Q$ is also absolutely continuous with respect to $P$, then the symmetrized KL divergence (a.k.a. Jeffrey's divergence [36]) is

$$D_{\text{SKL}}(P\|Q) \triangleq D(P\|Q) + D(Q\|P). \tag{4}$$

The mutual information between two random variables $X$ and $Y$ is defined as the KL divergence between the joint distribution and product-of-marginal distribution $I(X; Y) \triangleq D(P_{X,Y}\|P_X \otimes P_Y)$, or equivalently, the conditional KL divergence between $P_{Y|X}$ and $P_Y$ averaged over $P_X$, $D(P_{Y|X}\|P_Y|P_X) \triangleq \int_{\mathcal{X}} D(P_{Y|X=x}\|P_Y) dP_X(x)$. By swapping the role of $P_{X,Y}$ and $P_X \otimes P_Y$ in mutual information, we get the lautum information introduced by [49], $L(X; Y) \triangleq D(P_X \otimes P_Y\|P_{X,Y})$. Finally, the symmetrized KL information between $X$ and $Y$ is given by [6]:

$$I_{\text{SKL}}(X; Y) \triangleq D_{\text{SKL}}(P_{X,Y}\|P_X \otimes P_Y) = I(X; Y) + L(X; Y). \tag{5}$$

Throughout the paper, upper-case letters denote random variables (e.g., $Z$), lower-case letters denote the realizations of random variables (e.g., $z$), and calligraphic letters denote sets (e.g., $\mathcal{Z}$). All the logarithms are the natural ones, and all the information measure units are nats. $\mathcal{N}(\mu, \Sigma)$ denotes the Gaussian distribution with mean $\mu$ and covariance matrix $\Sigma$.

## 1.2 Contributions

The core contribution of this paper (see Theorem 1) is an *exact* characterization of the expected generalization error for the Gibbs algorithm in terms of the symmetrized KL information between the input training samples $S$ and the output hypothesis $W$, as follows:

$$\overline{\text{gen}}(P_{W|S}^\alpha, P_S) = \frac{I_{\text{SKL}}(W; S)}{\alpha}.$$

This result highlights the fundamental role of such an information quantity in learning theory that does not appear to have been recognized before. We also discuss some general properties of the symmetrized KL information, which could be used to prove the non-negativity and concavity of the expected generalization error for the Gibbs algorithm.

Building upon this result, we further expand our contributions in various directions:

- In Section 3, we tighten existing expected generalization error bound (see Theorem 2) and PAC-Bayesian bound (see Theorem 3) Gibbs algorithm under i.i.d and sub-Gaussian assumptions by combining our symmetrized KL information characterization with the existing bounding techniques.

- In Section 4 (Proposition 1 and 2), we show how to use our method to characterize the asymptotic behavior of the generalization error for Gibbs algorithm under large inverse temperature limit $\alpha \to \infty$, where Gibbs algorithm converges to the empirical risk minimization algorithm. Note that existing bounds, such as [39, 52, 71], become vacuous in this regime.

- In Section 5, we characterize the generalization error of the Gibbs algorithm with data-dependent regularizer using symmetrized KL information, which provides some insights on how to reduce the generalization error using regularization.

### 1.3 Motivations for the Gibbs Algorithm

As we discuss below, the choice of Gibbs algorithm is not arbitrary since it arises naturally in many different applications and is sufficiently general to model many learning algorithms used in practice:

**Empirical Risk Minimization:** The $(\alpha, \pi(w), L_E(w, s))$-Gibbs algorithm can be viewed as a randomized version of the empirical risk minimization (ERM) algorithm if we specify the energy function $f(w, s) = L_E(w, s)$. As the inverse temperature $\alpha \to \infty$, the prior distribution $\pi(w)$ becomes negligible, and the Gibbs algorithm converges to the standard ERM algorithm.

**Information Risk Minimization:** The Gibbs algorithm also arises when conditional KL-divergence is used as a regularizer to penalize over-fitting in the information risk minimization framework. In particular, it is shown in [71, 75, 76] that the solution to the following regularized ERM problem

$$P^\star_{W|S} = \arg\inf_{P_{W|S}} \left( \mathbb{E}_{P_{W,S}}[L_E(W, S)] + \frac{1}{\alpha} D(P_{W|S}\|\pi(W)|P_S) \right), \tag{6}$$

corresponds to the $(\alpha, \pi(w), L_E(w, s))$-Gibbs distribution. The inverse temperature $\alpha$ controls the regularization term and balances between over-fitting and generalization.

**PAC-Bayesian Bound:** The following upper bound on population risk from [63] holds with probability at least $1 - \delta$ for $0 < \delta < 1$, and $0 < \lambda < 2$ under distribution $P_S$,

$$\mathbb{E}_{P_{W|S=s}}[L_P(W, P_S)] \leq \frac{\mathbb{E}_{P_{W|S=s}}[L_E(W, s)]}{1 - \frac{\lambda}{2}} + \frac{D(P_{W|S=s}\|\pi(W)) + \log(\frac{2\sqrt{n}}{\delta})}{\lambda(1 - \frac{\lambda}{2})n}. \tag{7}$$

If we fix $\lambda, \pi(w)$ and optimize over $P_{W|S=s}$, the distribution that minimizes the PAC-Bayes bound in (7) would be the $(n\lambda, \pi(w), L_E(w, s))$-Gibbs distribution. Similar bounds are proposed in [21, Theorem 1.2.1] and [65, Lemma 10], where optimizing over posterior distribution would result in a Gibbs distribution.

**SGLD Algorithm:** The Stochastic Gradient Langevin Dynamics (SGLD) can be viewed as the discrete version of the continuous-time Langevin diffusion, and it is defined as follows:

$$W_{k+1} = W_k - \beta \nabla L_E(W_k, s) + \sqrt{\frac{2\beta}{\alpha}} \zeta_k, \quad k = 0, 1, \cdots, \tag{8}$$

where $\zeta_k$ is a standard Gaussian random vector and $\beta > 0$ is the step size. In [51], it is proved that under some conditions on loss function, the conditional distribution $P_{W_k|S}$ induced by SGLD algorithm is close to $(\alpha, \pi(W_0), L_E(w_k, s))$-Gibbs distribution in 2-Wasserstein distance for sufficiently large $k$. Under some conditions on the loss function $\ell(w, z)$, [22, 42] shows that in the continuous-time Langevin diffusion, the stationary distribution of hypothesis $W$ is the Gibbs distribution.

### 1.4 Other Related Work

**Information-theoretic generalization error bounds:** Recently, [58, 71] propose to use the mutual information between the input training set and the output hypothesis to upper bound the expected generalization error. However, those bounds are known not to be tight, and multiple approaches have been proposed to tighten the mutual information-based bound. [19] provides tighter bounds by considering the individual sample mutual information, [10, 11] propose using chaining mutual information, and [30, 31, 62] advocate the conditioning and processing techniques. Information-theoretic generalization error bounds using other information quantities are also studied, such as, $f$-divergence [37], $\alpha$-Rényi divergence and maximal leakage [25, 34], Jensen-Shannon divergence [7] and Wasserstein distance [41, 56, 69]. Using rate-distortion theory, [17, 18, 43] provide information-theoretic generalization error upper bounds for model misspecification and model compression.

**PAC-Bayesian generalization error bounds:** First proposed by [44, 45, 60], PAC-Bayesian analysis provides high probability bounds on the generalization error in terms of KL divergence between the data-dependent posterior induced by the learning algorithm and a data-free prior that can be chosen arbitrarily. There are multiple ways to generalize the standard PAC-Bayesian bounds, including using different information measures other than the KL divergence [3, 8, 14, 32, 48] and considering data-dependent priors (prior depends on the training data) [5, 13, 21, 23, 24, 53] or distribution-dependent priors (prior depends on data-generating distribution) [20, 40, 50, 54]. In [27], a more general PAC-Bayesian framework is proposed, which provides a high probability bound on the convex function of the expected population and empirical risk with respect to the posterior distribution, whereas in [26] the connection between Bayesian inference and PAC-Bayesian theorem is explored by considering Gibbs posterior and negative log loss function.

**Generalization error of Gibbs algorithm:** Both information-theoretic and PAC-Bayesian approaches have been used to bound the generalization error of the Gibbs algorithm. An information-theoretic upper bound with a convergence rate of $\mathcal{O}\left(\alpha/n\right)$ is provided in [52] for the Gibbs algorithm with bounded loss function, and PAC-Bayesian bounds using a variational approximation of Gibbs posteriors are studied in [4]. [11, Appendix D] provides an upper bound on the excess risk of the Gibbs algorithm under sub-Gaussian assumption. [39] focuses on the excess risk of the Gibbs algorithm and a similar generalization bound with rate of $\mathcal{O}\left(\alpha/n\right)$ is provided under sub-Gaussian assumption. Although these bounds are tight in terms of the sample complexity $n$, they become vacuous when the inverse temperature $\alpha \to \infty$, hence are unable to capture the behaviour of the ERM algorithm.

Our work differs from this body of research in the sense that we provide an exact characterization of generalization error of the Gibbs algorithm in terms of the symmetrized KL information. Our work also further leverages this characterization to tighten the existing expected and PAC-Bayesian generalization error bounds in literature.

## 2  Generalization Error of Gibbs Algorithm

Our main result, which characterizes the exact expected generalization error of the Gibbs algorithm with prior distribution $\pi(w)$, is as follows:

**Theorem 1.** *For* $(\alpha, \pi(w), L_E(w,s))$*-Gibbs algorithm,*

$$P_{W|S}^\alpha(w|s) = \frac{\pi(w)e^{-\alpha L_E(w,s)}}{V(s,\alpha)}, \quad \alpha > 0, \tag{9}$$

*the expected generalization error is given by*

$$\overline{gen}(P_{W|S}^\alpha, P_S) = \frac{I_{\mathrm{SKL}}(W;S)}{\alpha}. \tag{10}$$

*Sketch of Proof:* It can be shown that the symmetrized KL information can be written as

$$I_{\mathrm{SKL}}(W;S) = \mathbb{E}_{P_{W,S}}[\log(P_{W|S}^\alpha)] - \mathbb{E}_{P_W \otimes P_S}[\log(P_{W|S}^\alpha)]. \tag{11}$$

Just like the generalization error, the above expression is the difference between the expectations of the same function evaluated under the joint distribution and the product-of-marginal distribution. Note that $P_{W,S}$ and $P_W \otimes P_S$ share the same marginal distribution, we have $\mathbb{E}_{P_{W,S}}[\log \pi(W)] =$

$\mathbb{E}_{P_W}[\log \pi(W)]$, and $\mathbb{E}_{P_{W,S}}[\log V(S, \alpha)] = \mathbb{E}_{P_S}[\log V(S, \alpha)]$. Then, combining (9) with (11) completes the proof. More details together with the full proof are provided in Appendix B.1. $\quad\square$

To the best of our knowledge, this is the first exact characterization of the expected generalization error for the Gibbs algorithm. Note that Theorem 1 only assumes that the loss function is non-negative, and it holds even for non-i.i.d training samples.

In Section 2.1, we discuss some general properties of the expected generalization error that can be learned directly from the properties of symmetrized KL information. In Section 2.2, we provide a mean estimation example to show that the symmetrized KL information can be computed exactly for squared loss with Gaussian prior.

## 2.1 General Properties

By Theorem 1, some basic properties of the expected generalization error, including non-negativity and concavity, can be proved directly from the properties of symmetrized KL information.

The non-negativity of the expected generalization error, i.e., $\overline{\mathrm{gen}}(P^\alpha_{W|S}, P_S) \geq 0$, follows by the non-negativity of the symmetrized KL information. Note that the non-negativity result could also be proved using [39, Appendix A.2] under much more stringent assumptions, including i.i.d samples and a sub-Gaussian loss function.

It is shown in [6] that the symmetrized KL information $I_{\mathrm{SKL}}(X; Y)$ is a concave function of $P_X$ for fixed $P_{Y|X}$, and a convex function of $P_{Y|X}$ for fixed $P_X$. Thus, we have the following corollary.

**Corollary 1.** *For a fixed* $(\alpha, \pi(w), L_E(w, s))$*-Gibbs algorithm* $P^\alpha_{W|S}$*, the expected generalization error* $\overline{\mathrm{gen}}(P^\alpha_{W|S}, P_S)$ *is a concave function of* $P_S$*.*

The concavity of the generalization error for the Gibbs algorithm $P^\alpha_{W|S}$ can be immediately used to explain why training a model by mixing multiple datasets from different domains leads to poor generalization. Suppose that the data-generating distribution is domain-dependent, i.e., there exists a random variable $D$, such that $D \leftrightarrow S \leftrightarrow W$ holds. Then, $P_S = \mathbb{E}_{P_D}[P_{S|D}]$ can be viewed as the mixture of the data-generating distribution across all domains. From Corollary 1 and Jensen's inequality, we have

$$\overline{\mathrm{gen}}(P^\alpha_{W|S}, P_S) \geq \mathbb{E}_{P_D}\left[\overline{\mathrm{gen}}(P^\alpha_{W|S}, P_{S|D})\right], \tag{12}$$

which shows that the generalization error of Gibbs algorithm achieved with the mixture distribution $P_S$ is larger than the averaged generalization error for each $P_{S|D}$.

More discussions about other properties of symmetrized KL divergence, including data processing inequality ( symmetrized KL divergence is an $f$-divergence), variational representation, chain rule, and their implications in learning problems are provided in Appendix B.2.

## 2.2 Example: Mean Estimation

We now consider a simple learning problem, where the symmetrized KL information can be computed exactly, to demonstrate the usefulness of Theorem 1. All details are provided in Appendix B.3.

Consider the problem of learning the mean $\boldsymbol{\mu} \in \mathbb{R}^d$ of a random vector $Z$ using $n$ i.i.d training samples $S = \{Z_i\}_{i=1}^n$. We assume that the covariance matrix of $Z$ satisfies $\Sigma_Z = \sigma_Z^2 I_d$ with unknown $\sigma_Z^2$. We adopt the mean-squared loss $\ell(\boldsymbol{w}, \boldsymbol{z}) = \|\boldsymbol{z} - \boldsymbol{w}\|_2^2$, and assume a Gaussian prior for the mean $\pi(\boldsymbol{w}) = \mathcal{N}(\boldsymbol{\mu}_0, \sigma_0^2 I_d)$. If we set inverse-temperature $\alpha = \frac{n}{2\sigma^2}$, then the $(\frac{n}{2\sigma^2}, \mathcal{N}(\boldsymbol{\mu}_0, \sigma_0^2 I_d), L_E(\boldsymbol{w}, s))$-Gibbs algorithm is given by the following posterior distribution [47],

$$P^\alpha_{W|S}(\boldsymbol{w}|Z^n) \sim \mathcal{N}\Big(\frac{\sigma_1^2}{\sigma_0^2}\boldsymbol{\mu}_0 + \frac{\sigma_1^2}{\sigma^2}\sum_{i=1}^n Z_i, \sigma_1^2 I_d\Big), \quad \text{with} \quad \sigma_1^2 = \frac{\sigma_0^2 \sigma^2}{n\sigma_0^2 + \sigma^2}. \tag{13}$$

Since $P^\alpha_{W|S}$ is Gaussian, the mutual information and lautum information are given by

$$I(S; W) = \frac{nd\sigma_0^2 \sigma_Z^2}{2(n\sigma_0^2 + \sigma^2)\sigma^2} - D\big(P_W \| \mathcal{N}(\boldsymbol{\mu}_W, \sigma_1^2 I_d)\big), \tag{14}$$

$$L(S; W) = \frac{nd\sigma_0^2 \sigma_Z^2}{2(n\sigma_0^2 + \sigma^2)\sigma^2} + D\big(P_W \| \mathcal{N}(\boldsymbol{\mu}_W, \sigma_1^2 I_d)\big), \quad \text{with} \quad \boldsymbol{\mu}_W = \frac{\sigma_1^2}{\sigma_0^2}\boldsymbol{\mu}_0 + \frac{n\sigma_1^2}{\sigma^2}\boldsymbol{\mu}. \tag{15}$$

For additive Gaussian channel $P_{W|S}$, it is well known that Gaussian input distribution (which also gives a Gaussian output distribution $P_W$) maximizes the mutual information under a second-order moment constraint. As we can see from the above expressions, the opposite is true for lautum information. In addition, symmetrized KL information $I_{\text{SKL}}(W; S)$ is independent of the distribution of $P_Z$, as long as $\Sigma_Z = \sigma_Z^2 I_d$.

From Theorem 1, the generalization error of this algorithm can be computed exactly as:

$$\overline{\text{gen}}(P_{W|S}^\alpha, P_S) = \frac{I_{\text{SKL}}(W; S)}{\alpha} = \frac{2d\sigma_0^2\sigma_Z^2}{n\sigma_0^2 + \sigma^2} = \frac{2d\sigma_0^2\sigma_Z^2}{n(\sigma_0^2 + \frac{1}{2\alpha})}, \tag{16}$$

which has the decay rate of $\mathcal{O}(1/n)$. As a comparison, the individual sample mutual information (ISMI) bound from [19], which is shown to be tighter than the mutual information-based bound in [71, Theorem 1], gives a sub-optimal bound with order $\mathcal{O}(1/\sqrt{n})$, as $n \to \infty$, (see Appendix B.4).

## 3 Tighter Generalization Error Upper Bounds

In this section, we show that by combining Theorem 1 with existing information-theoretic and PAC-Bayesian approaches, we can provide tighter generalization error upper bounds for the Gibbs algorithm. These bounds quantify how the generalization error of the Gibbs algorithm depends on the number of samples $n$, and are useful when directly evaluating the symmetrized KL information is hard.

### 3.1 Expected Generalization Error Upper Bound

The following upper bound on the expected generalization error for the Gibbs algorithm can be obtained by combining our Theorem 1 with the information-theoretic bound proposed in [71] under i.i.d and sub-Gaussian assumptions.

**Theorem 2.** *(proved in Appendix C.1) Suppose that the training samples $S = \{Z_i\}_{i=1}^n$ are i.i.d generated from the distribution $P_Z$, and the non-negative loss function $\ell(w, Z)$ is $\sigma$-sub-Gaussian on the left-tail* [*] under distribution $P_Z$ for all $w \in \mathcal{W}$. If we further assume $C_E \leq \frac{L(W;S)}{I(W;S)}$ for some $C_E \geq 0$, then for the $(\alpha, \pi(w), L_E(w, s))$-Gibbs algorithm, we have*

$$0 \leq \overline{\text{gen}}(P_{W|S}^\alpha, P_S) \leq \frac{2\sigma^2\alpha}{(1 + C_E)n}. \tag{17}$$

Theorem 2 establishes the convergence rate $\mathcal{O}(\alpha/n)$ of the generalization error of Gibbs algorithm with i.i.d training samples, and suggests that a smaller inverse temperature $\alpha$ leads to a tighter upper bound. Note that all the $\sigma$-sub-Gaussian loss functions are also $\sigma$-sub-Gaussian on the left-tail under the same distribution (loss function in Section 2.2 is sub-Gaussian on the left-tail under $P_Z$, but not sub-Gaussian). Therefore, our result also applies to any bounded loss function $\ell : \mathcal{W} \times \mathcal{Z} \to [a, b]$, since bounded functions are $(\frac{b-a}{2})$-sub-Gaussian.

**Remark 1** (Previous Results). *Using the fact that Gibbs algorithm is differentially private [46] for bounded loss functions $\ell \in [0, 1]$, directly applying [71, Theorem 1] gives a sub-optimal bound $|\overline{\text{gen}}(P_{W|S}^\alpha, P_S)| \leq \sqrt{\frac{\alpha}{n}}$. By further exploring the bounded loss assumption using Hoeffding's lemma, a tighter upper bound $|\overline{\text{gen}}(P_{W|S}^\alpha, P_S)| \leq \frac{\alpha}{2n}$ is obtained in [52], which has the similar decay rate order of $\mathcal{O}(\alpha/n)$. In [39, Theorem 1], the upper bound $\overline{\text{gen}}(P_{W|S}^\alpha, P_S) \leq \frac{4\sigma^2\alpha}{n}$ is derived with a different assumption, i.e., $\ell(W, z)$ is $\sigma$-sub-Gaussian under Gibbs algorithm $P_{W|S}^\alpha$. In Theorem 2, we assume the loss function is $\sigma$-sub-Gaussian on left-tail under data-generating distribution $P_Z$ for all $w \in \mathcal{W}$, which is more general as we discussed above. Our upper bound is also improved by a factor of $\frac{1}{2(1+C_E)}$ compared to the result in [39].*

**Remark 2** (Choice of $C_E$). *Since $L(W; S) > 0$ when $I(W; S) > 0$, setting $C_E = 0$ is always valid in Theorem 2, which gives $\overline{\text{gen}}(P_{W|S}^\alpha, P_S) \leq \frac{2\sigma^2\alpha}{n}$. As shown in [49, Theorem 15], $L(S; W) \geq$*

---

[*]A random variable $X$ is $\sigma$-sub-Gaussian if $\log \mathbb{E}[e^{\lambda(X - \mathbb{E}X)}] \leq \frac{\sigma^2\lambda^2}{2}, \forall\lambda \in \mathbb{R}$, and $X$ is $\sigma$-sub-Gaussian on the left-tail if $\log \mathbb{E}[e^{\lambda(X - \mathbb{E}X)}] \leq \frac{\sigma^2\lambda^2}{2}, \forall\lambda \leq 0$.

$I(S;W)$ holds for any Gaussian channel $P_{W|S}$. In addition, it is discussed in [49, Example 1], if either the entropy of training $S$ or the hypothesis $W$ is small, $I(S;W)$ would be smaller than $L(S;W)$ (as it is not upper-bounded by the entropy), which implies that the lautum information term is not negligible in general.

We extend Theorem 2 by considering other types of tail behavior including sub-Gamma and sub-Exponential on left-tail in Appendix C.2.

## 3.2 PAC-Bayesian Upper Bound

As discussed in Section 1.4, the prior distribution used in PAC-Bayesian bounds is different from the prior in Gibbs algorithm, since the former priors can be chosen arbitrarily to tighten the generalization error bound. In this section, we provide a tighter PAC-Bayesian bound based on the symmetrized KL divergence, which is inspired by the distribution-dependent PAC-Bayesian bound proposed in [40] using $(\alpha, \pi(w), L_P(w, P_S))$-Gibbs distribution as the PAC-Bayesian prior.

As the data-generating distribution $P_S$ is unknown in practice, we consider the $(\alpha, \pi(w), L_P(w, P_{S'}))$-Gibbs distribution in the following discussion, where $P_{S'}$ is an arbitrary data-generating distribution. Since $(\alpha, \pi(w), L_P(w, P_{S'}))$-Gibbs distribution is independent of the samples $S$ and only depends on the population risk $L_P(w, P_{S'})$, we can denote it as $P_W^{\alpha, L'_P}$.

By exploiting the connection between the symmetrized KL divergence $D_{\mathrm{SKL}}(P_{W|S=s}^\alpha \| P_W^{\alpha, L'_P})$ and the KL divergence term $D(P_{W|S=s}^\alpha \| P_W^{\alpha, L'_P})$ in the PAC-Bayesian bound from [40], the following PAC-Bayesian bound can be obtained under i.i.d and sub-Gaussian assumptions.

**Theorem 3.** *(proved in Appendix D) Suppose that the training samples $S = \{Z_i\}_{i=1}^n$ are i.i.d generated from the distribution $P_Z$, and the non-negative loss function $\ell(w, Z)$ is $\sigma$-sub-Gaussian under data-generating distribution $P_Z$ for all $w \in \mathcal{W}$. If we use the $(\alpha, \pi(w), L_P(w, P_{S'}))$-Gibbs distribution as the PAC-Bayesian prior, where $P_{S'}$ is an arbitrary chosen (and known) distribution, the following upper bound holds for the generalization error of $(\alpha, \pi(w), L_E(w, s))$-Gibbs algorithm with probability at least $1 - 2\delta$, $0 < \delta < \frac{1}{2}$ under distribution $P_S$,*

$$\left| \mathbb{E}_{P_{W|S=s}^\alpha}[L_P(W, P_S) - L_E(W, s)] \right| \leq \frac{2\sigma^2 \alpha}{(1 + C_P(s))n} + 2\sqrt{\frac{\sigma^2 \alpha}{(1 + C_P(s))n}} \left( \sqrt[4]{2\sigma^2 D(P_{Z'} \| P_Z)} + \epsilon \right) + \epsilon^2,$$

*where $\epsilon \triangleq \sqrt[4]{\frac{2\sigma^2 \log(1/\delta)}{n}}$, and $C_P(s) \leq \frac{D\left(P_W^{\alpha, L'_P} \big\| P_{W|S=s}^\alpha\right)}{D\left(P_{W|S=s}^\alpha \big\| P_W^{\alpha, L'_P}\right)}$ for some $C_P(s) \geq 0$.*

**Remark 3** (Previous Result). *We could recover the distribution-dependent bound in [40, Theorem 6] by setting $P_{Z'} = P_Z$, choosing a bounded loss function in $[0, 1]$ and $C_P(s) = 0$ in our Theorem 3. Note that multiple terms in our upper bound in Theorem 3 are tightened by a factor of $1/(1 + C_P(s))$, and we also consider $\sigma$-sub-Gaussian loss functions.*

**Remark 4** (Choice of $C_P(s)$). *Since the distribution $P_{Z'}$ can be set arbitrarily, the prior distribution $P_W^{\alpha, L'_P}$ is accessible. Then, we can set $C_P(s) = D\left(P_W^{\alpha, L'_P} \big\| P_{W|S=s}^\alpha\right) / D\left(P_{W|S=s}^\alpha \big\| P_W^{\alpha, L'_P}\right)$ to tighten the bound, as it can be computed exactly using the training set.*

## 4  Asymptotic Behavior of Generalization Error for Gibbs Algorithm

In this section, we consider the asymptotic behavior of the generalization error for Gibbs algorithm as the inverse temperature $\alpha \to \infty$. Note that the upper bounds obtained in the previous section, as well as the ones in the literature, have the order $\mathcal{O}(\frac{\alpha}{n})$, which becomes vacuous in this regime. However, it is known that the Gibbs algorithm will converge to ERM as $\alpha \to \infty$, which has finite generalization error with bounded loss function. To resolve this issue, we provide an exact characterization of the generalization error in this regime using Theorem 1.

It is shown in [12, 33] that the asymptotic behavior of the Gibbs algorithm depends on the number of minimizers for the empirical risk, so we consider the single-well case and multiple-well case separately.

**Single-well case:** In this case, there exists a unique $W^*(S)$ that minimizes the empirical risk, i.e.,

$$W^*(S) = \underset{w \in \mathcal{W}}{\arg\min} \, L_E(w, S). \tag{18}$$

It is shown in [33] that if $H^*(S) \triangleq \nabla_w^2 L_E(w, S)\big|_{w=W^*(S)}$ is not singular, then $P_{W|S}^\alpha \to \mathcal{N}(W^*(S), \frac{1}{\alpha} H^*(S)^{-1})$ in distribution. Thus, the symmetrized KL information in Theorem 1 can be evaluated using this Gaussian approximation, which gives the following result.

**Proposition 1.** *(proved in Appendix E.1) In the single-well case, if the Hessian matrix $H^*(S)$ is not singular, then the generalization error of the $(\infty, \pi(\boldsymbol{w}), L_E(\boldsymbol{w}, s))$-Gibbs algorithm is*

$$\overline{gen}(P_{W|S}^\infty, P_S) = \mathbb{E}_{\Delta_{W,S}}\left[\frac{1}{2} W^\top H^*(S) W\right] \tag{19}$$
$$+ \mathbb{E}_{P_S}\left[(W^*(S) - \mathbb{E}[W^*(S)])^\top (H^*(S)W^*(S) - \mathbb{E}[H^*(S)W^*(S)])\right],$$

*where $\mathbb{E}_{\Delta_{W,S}}[f(W, S)] \triangleq \mathbb{E}_{P_W \otimes P_S}[f(W, S)] - \mathbb{E}_{P_{W,S}}[f(W, S)]$.*

Proposition 1 shows that the generalization error of the Gibbs algorithm in the limiting regime $\alpha \to \infty$ highly depends on the landscape of the empirical risk function.

As an example, we use Proposition 1 to obtain the generalization error of the maximum likelihood estimates (MLE) in the asymptotic regime $n \to \infty$. More specifically, suppose that we have $n$ i.i.d. training samples generated from the distribution $P_Z$, and we want to fit the training data with a parametric distribution family $\{f(z_i|\boldsymbol{w})\}_{i=1}^n$, where $\boldsymbol{w} \in \mathcal{W} \subset \mathbb{R}^d$ denotes the parameter. Here, the true data-generating distribution may not belong to the parametric family, i.e., $P_Z \neq f(\cdot|\boldsymbol{w})$ for $\boldsymbol{w} \in \mathcal{W}$. If we use the log-loss $\ell(\boldsymbol{w}, z) = -\log f(z|\boldsymbol{w})$ in the Gibbs algorithm, as $\alpha \to \infty$, it converges to the ERM algorithm, which is equivalent to MLE, i.e.,

$$W^*(S) = \hat{W}_{\text{ML}} \triangleq \arg\max_{\boldsymbol{w} \in \mathcal{W}} \sum_{i=1}^n \log f(Z_i|\boldsymbol{w}). \tag{20}$$

As $n \to \infty$, under regularization conditions (details in Appendix E.2) which guarantee that $W^*(S)$ is unique, the asymptotic normality of the MLE [64] states that the distribution of $\hat{W}_{\text{ML}}$ converges to

$$\mathcal{N}(\boldsymbol{w}^*, \frac{1}{n} J(\boldsymbol{w}^*)^{-1} \mathcal{I}(\boldsymbol{w}^*) J(\boldsymbol{w}^*)^{-1}), \quad \text{with} \quad \boldsymbol{w}^* \triangleq \arg\min_{\boldsymbol{w} \in \mathcal{W}} D(P_Z \| f(\cdot|\boldsymbol{w})),$$

$$J(\boldsymbol{w}) \triangleq \mathbb{E}_Z\left[-\nabla_{\boldsymbol{w}}^2 \log f(Z|\boldsymbol{w})\right] \quad \text{and} \quad \mathcal{I}(\boldsymbol{w}) \triangleq \mathbb{E}_Z\left[\nabla_{\boldsymbol{w}} \log f(Z|\boldsymbol{w}) \nabla_{\boldsymbol{w}} \log f(Z|\boldsymbol{w})^\top\right].$$

In addition, the Hessian matrix $H^*(S) \to J(\boldsymbol{w}^*)$ as $n \to \infty$, which is independent of the training samples $S$. Thus, $\mathbb{E}_{\Delta_{W,S}}[\frac{1}{2} W^\top H^*(S) W] = 0$, and Proposition 1 gives

$$\overline{gen}(P_{W|S}^\infty, P_S) = \frac{\text{tr}(\mathcal{I}(\boldsymbol{w}^*) J(\boldsymbol{w}^*)^{-1})}{n}. \tag{21}$$

When the true model is in the parametric family $P_Z = f(\cdot|\boldsymbol{w}^*)$, we have $\mathcal{I}(\boldsymbol{w}^*) = J(\boldsymbol{w}^*)$ and the above expression reduces to $\overline{gen}(P_{W|S}^\infty, P_Z) = \frac{d}{n}$, which corresponds to the well-known Akaike information criterion (AIC) [2] used in MLE model selection.

**Multiple-well case:** In this case, there exist $M$ distinct $W_u^*(S)$ such that
$$W_u^*(S) \in \arg\min_{w \in \mathcal{W}} L_E(w, S), \quad u \in \{1, \cdots, M\}, \tag{22}$$
where $M$ is a fixed constant, and all the minimizers $W_u^*(S)$ are isolated, meaning that a sufficiently small neighborhood of each $W_u^*(S)$ contains a unique minimum.

In this multiple-well case, it is shown in [12] that the the Gibbs algorithm can be approximated by a Gaussian mixture, as long as $H_u^*(S) \triangleq \nabla_w^2 L_E(w, S)\big|_{w=W_u^*(S)}$ is not singular for all $u \in \{1, \cdots, M\}$. However, there is no closed form for the symmetrized KL information for Gaussian mixtures. Thus, we provide the following upper bound of the generalization error by evaluating Theorem 1 under the assumption that $\pi(W)$ is a uniform distribution over $\mathcal{W}$.

**Proposition 2.** *(proved in Appendix E.1) If we assume that $\pi(W)$ is a uniform distribution over $\mathcal{W}$, and the Hessian matrices $H_u^*(S)$ are not singular for all $u \in \{1, \cdots, M\}$, then the generalization error of the $(\infty, \pi(\boldsymbol{w}), L_E(\boldsymbol{w}, s))$-Gibbs algorithm in the multiple-well case can be bounded as*

$$\overline{gen}(P_{W|S}^\infty, P_S) \leq \frac{1}{M} \sum_{u=1}^M \left[\mathbb{E}_{\Delta_{W_u,S}}\left[\frac{1}{2} W_u^\top H_u^*(S) W_u\right]\right.$$
$$\left. + \mathbb{E}_{P_S}\left[(W_u^*(S) - \mathbb{E}[W_u^*(S)])^\top H_u(W_u^*(S) - \mathbb{E}[W_u^*(S)])\right]\right]. \tag{23}$$

Comparing with Proposition 1, Proposition 2 shows that the global generalization error in the multiple-well case can be upper bounded by the mean of the generalization errors achieved by each local minimizer.

**Remark 5.** *In [39], a similar Gaussian approximation technique is used to bound the* excess risk *of Gibbs algorithm in both single-well and multiple-well cases. However, their result is based on a loose generalization error bound with the order $\mathcal{O}(\frac{\alpha}{n})$. Thus, our method can also be used to obtain a tighter characterization of the excess risk for the Gibbs algorithm.*

In Appendix E.3, we consider a slightly different asymptotic regime, where the Gibbs algorithm converges to the Bayesian posterior instead of ERM. A similar result as in (21) can be obtained from Bernstein–von–Mises theorem [38] and the asymptotic normality of the MLE.

## 5 Regularized Gibbs Algorithm

In this section, we show how regularization will influence the generalization error of the Gibbs algorithm. Our definition of the regularizer is more general compared to the standard data-independent regularizer, as it may also depend on the training samples. There are many applications of such data-dependent regularization in the literature—e.g., data-dependent spectral norm regularization is proposed in [57], $\ell_1$ regularizer over data-dependent hypothesis space is studied in [70] and dropout is modeled as data-dependent $\ell_2$ regularization in [68].

In the following proposition, we consider the Gibbs algorithm with a regularization term $R : \mathcal{W} \times \mathcal{Z}^n \to \mathbb{R}_0^+$ and characterize the generalization error of this $(\alpha, \pi(w), L_E(w, s) + \lambda R(w, s))$-Gibbs algorithm, which is the solution of the following regularized ERM problem:

$$P_{W|S}^\star = \arg \inf_{P_{W|S}} \left( \mathbb{E}_{P_{W,S}}[L_E(W, S) + \lambda R(W, S)] + \frac{1}{\alpha} D(P_{W|S} \| \pi(W) | P_S) \right), \quad (24)$$

where $\lambda \geq 0$ controls the regularization term.

**Proposition 3.** *(proved in Appendix F.1) For $(\alpha, \pi(w), L_E(w, s) + \lambda R(w, s))$-Gibbs algorithm, its expected generalization error is given by*

$$\overline{gen}(P_{W|S}^\alpha, P_S) = \frac{I_{\mathrm{SKL}}(W; S)}{\alpha} - \lambda \mathbb{E}_{\Delta_{W,S}}[R(W, S)], \quad (25)$$

*where $\mathbb{E}_{\Delta_{W,S}}[R(W, S)] = \mathbb{E}_{P_W \otimes P_S}[R(W, S)] - \mathbb{E}_{P_{W,S}}[R(W, S)]$.*

Proposition 3 holds for non-i.i.d samples and any non-negative loss function, and it shows that to improve the generalization ability of the Gibbs algorithm, the data-dependent regularizer needs to 1) minimize the symmetrized KL information $I_{\mathrm{SKL}}(W; S)$ and 2) maximize the $\mathbb{E}_{\Delta_{W,S}}[R(W, S)]$ term which corresponds to a "generalization error" defined with the regularization term $R(W, S)$.

**Remark 6.** *If the regularizer is independent of the data, i.e., $R(w, s) = R(w)$, we have $\mathbb{E}_{\Delta_{W,S}}[R(W, S)] = 0$, and Proposition 3 gives $\overline{gen}(P_{W|S}^\alpha, P_S) = \frac{I_{\mathrm{SKL}}(W; S)}{\alpha}$, which implies that the data-independent regularizer needs to improve the generalization ability of learning algorithm by reducing the symmetrized KL information $I_{\mathrm{SKL}}(W; S)$.*

Inspired by the data-dependent regularizer proposed in [61] for support vector machines, we consider a similar data-dependent $\ell_2$-regularizer in the following proposition.

**Proposition 4.** *(proved in Appendix F.1) Suppose that we adopt the $\ell_2$-regularizer $R(w, s) = \|w - T(s)\|_2^2$, where $T(\cdot)$ is an arbitrary deterministic function $T : \mathcal{Z}^n \to \mathcal{W}$. Then, the expected generalization error of $(\alpha, \pi(w), L_E(w, s) + \lambda R(w, s))$-Gibbs algorithm is*

$$\overline{gen}(P_{W|S}^\alpha, P_S) = \frac{I_{\mathrm{SKL}}(W; S)}{\alpha} - \lambda \mathrm{tr}\big(\mathrm{Cov}[W, T(S)]\big), \quad (26)$$

*where $\mathrm{Cov}[W, T(S)]$ denotes the covariance matrix between $W$ and $T(S)$.*

Our result suggests that to reduce the generalization error with data-dependent $\ell_2$-regularizer, the function $T(S)$ should be chosen in a way, such that the term $\mathrm{tr}(\mathrm{Cov}[W, T(S)])$ is maximized. One way is to leave a part of the training set and learn the $T(S)$ function. Note that similar idea has been explored in the development of PAC-Bayesian bound with data-dependent prior [5]. More discussions and results about data-dependent regularizer are provided in Appendix F.2.

# 6 Conclusion

We provide an exact characterization of the generalization error for the Gibbs algorithm using symmetrized KL information. We demonstrate the power and versatility of our approach in multiple applications, including tightening expected generalization error and PAC-Bayesian bounds, characterizing the behaviors of the Gibbs algorithm with large inverse temperature and the regularized Gibbs algorithm.

This work motivates further investigation of the Gibbs algorithm in a variety of settings, including extending our results to characterize the generalization ability of an over-parameterized Gibbs algorithm, which could potentially provide more understanding of the generalization ability for deep learning.

## Acknowledgments and Disclosure of Funding

We thank the anonymous reviewers for their valuable feedback, which helped us to improve the paper greatly. Gholamali Aminian is supported by the Royal Society Newton International Fellowship, grant no. NIF\R1 \192656. Yuheng Bu is supported, in part, by NSF under Grant CCF-1717610 and by the MIT-IBM Watson AI Lab.

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
