# An Exact Characterization of the Generalization Error for the Gibbs Algorithm: Supplementary Material

**Gholamali Aminian** *
University College London
g.aminian @ucl.ac.uk

**Yuheng Bu** *
Massachusetts Institute of Technology
buyuheng@mit.edu

**Laura Toni**
University College London
l.toni@ucl.ac.uk

**Miguel Rodrigues**
University College London
m.rodrigues@ucl.ac.uk

**Gregory Wornell**
Massachusetts Institute of Technology
gww@mit.edu

## A Preliminaries

In this section, we introduce the notion of cumulant generating function, which characterizes different tail behaviors of random variables.

**Definition 1.** *The cumulant generating function (CGF) of a random variable $X$ is defined as*

$$\Lambda_X(\lambda) \triangleq \log \mathbb{E}[e^{\lambda(X-\mathbb{E}X)}]. \tag{27}$$

Assuming $\Lambda_X(\lambda)$ exists, it can be verified that $\Lambda_X(0) = \Lambda'_X(0) = 0$, and that it is convex.

**Definition 2.** *For a convex function $\psi$ defined on the interval $[0, b)$, where $0 < b \leq \infty$, its Legendre dual $\psi^\star$ is defined as*

$$\psi^\star(x) \triangleq \sup_{\lambda \in [0,b)} \left( \lambda x - \psi(\lambda) \right). \tag{28}$$

The following lemma characterizes a useful property of the Legendre dual and its inverse function.

**Lemma 1.** *[15, Lemma 2.4] Assume that $\psi(0) = \psi'(0) = 0$. Then $\psi^\star(x)$ defined above is a non-negative convex and non-decreasing function on $[0, \infty)$ with $\psi^\star(0) = 0$. Moreover, its inverse function $\psi^{\star-1}(y) = \inf\{x \geq 0 : \psi^\star(x) \geq y\}$ is concave, and can be written as*

$$\psi^{\star-1}(y) = \inf_{\lambda \in [0,b)} \left( \frac{y + \psi(\lambda)}{\lambda} \right), \quad b > 0. \tag{29}$$

We consider the distributions with the following tail behaviors in the appendices:

- **Sub-Gaussian:** A random variable $X$ is $\sigma$-sub-Gaussian, if $\psi(\lambda) = \frac{\sigma^2 \lambda^2}{2}$ is an upper bound on $\Lambda_X(\lambda)$, for $\lambda \in \mathbb{R}$. Then by Lemma 1,

$$\psi^{\star-1}(y) = \sqrt{2\sigma^2 y}.$$

- **Sub-Exponential:** A random variable $X$ is $(\sigma_e^2, b)$-sub-Exponential, if $\psi(\lambda) = \frac{\sigma_e^2 \lambda^2}{2}$ is an upper bound on $\Lambda_X(\lambda)$, for $0 \leq |\lambda| \leq \frac{1}{b}$ and $b > 0$. Using Lemma 1, we have

$$\psi^{\star-1}(y) = \begin{cases} \sqrt{2\sigma_e^2 y}, & \text{if } y \leq \frac{\sigma_e^2}{2b}; \\ by + \frac{\sigma_e^2}{2b}, & \text{otherwise}. \end{cases}$$

- **Sub-Gamma:** A random variable $X$ is $\Gamma(\sigma_s^2, c_s)$-sub-Gamma [74], if $\psi(\lambda) = \frac{\lambda^2 \sigma_s^2}{2(1 - c_s |\lambda|)}$ is an upper bound on $\Lambda_X(\lambda)$, for $0 < |\lambda| < \frac{1}{c_s}$ and $c_s > 0$. Using Lemma 1, we have

$$\psi^{\star -1}(y) = \sqrt{2\sigma_s^2 y} + c_s y.$$

Sub-Exponential condition is a slightly milder compared with sub-Gaussian condition. All the definition above can be generalized by considering only the left ($\lambda < 0$) or right ($\lambda > 0$) tails, e.g., $\sigma$-sub-Gaussian in the left tail as in Theorem 2.

## B  Generalization Error of Gibbs Algorithm

### B.1  Theorem 1 Details

We start with the following two Lemmas:

**Lemma 2.** *We define the following $J_E(w, S)$ function as a proxy for the empirical risk, i.e., $J_E(w, S) \triangleq \frac{\alpha}{n} \sum_{i=1}^{n} \ell(w, Z_i) + g(w) + h(S)$, where $\alpha \in \mathbb{R}_0^+$, $g : \mathcal{W} \to \mathbb{R}$, $h : \mathcal{Z}^n \to \mathbb{R}$, and the function $J_P(w, \mu) \triangleq \mathbb{E}_{P_S}[J_E(w, S)]$ as a proxy for the population risk. Then,*

$$\mathbb{E}_{P_{W,S}}[J_P(W, \mu) - J_E(W, S)] = \alpha \cdot \overline{gen}(P_{W|S}, P_S). \tag{30}$$

*Proof.*

$$
\begin{aligned}
&\mathbb{E}_{P_{W,S}}[J_P(W, \mu) - J_E(W, S)] \\
&= \mathbb{E}_{P_{W,S}} \Big[ \mathbb{E}_{P_{Z^n}} [\frac{\alpha}{n} \sum_{i=1}^{n} \ell(W, Z_i)] - \frac{\alpha}{n} \sum_{i=1}^{n} \ell(W, Z_i) \Big] \\
&\quad + \mathbb{E}_{P_W} \Big[ g(W) + \mathbb{E}_{P_S}[h(S)] \Big] - \mathbb{E}_{P_{W,S}} \Big[ g(W) + h(S) \Big] \\
&= \alpha \cdot \mathbb{E}_{P_{W,S}}[L_P(W, \mu) - L_E(W, S)] \\
&= \alpha \cdot \overline{gen}(P_{W|S}, P_S). \quad \square
\end{aligned}
\tag{31}
$$

**Lemma 3.** *Consider a learning algorithm $P_{W|S}$, if we set the proxy function $J_E(w, z^n) = -\log P_{W|S}(w|s)$, then*

$$\mathbb{E}_{P_{W,S}}[J_P(W, \mu) - J_E(W, S)] = I_{\mathrm{SKL}}(W; S). \tag{32}$$

*Proof.*

$$
\begin{aligned}
&I(W; S) + L(W; S) \\
&= \mathbb{E}_{P_{W,S}} \Big[ \log \frac{P_{W|S}(W|S)}{P_W(W)} \Big] + \mathbb{E}_{P_W \otimes P_S} \Big[ \log \frac{P_W(W)}{P_{W|S}(W|S)} \Big] \\
&= \mathbb{E}_{P_{W,S}} \Big[ \log P_{W|S}(W|S) \Big] - \mathbb{E}_{P_W \otimes P_S} \Big[ \log P_{W|S}(W|S) \Big] \\
&= \mathbb{E}_{P_{W,S}} [-\mathbb{E}_{P_S}[\log P_{W|S}(W|S)] + \log P_{W|S}(W|S)] \\
&= \mathbb{E}_{P_{W,S}}[J_P(W, \mu) - J_E(W, S)]. \quad \square
\end{aligned}
\tag{33}
$$

**Theorem 1.** *(restated) For $(\alpha, \pi(w), L_E(w, s))$-Gibbs algorithm,*

$$P_{W|S}^{\alpha}(w|s) = \frac{\pi(w) e^{-\alpha L_E(w,s)}}{V(s, \alpha)}, \quad \alpha > 0,$$

*its expected generalization error is given by*

$$\overline{gen}(P_{W|S}^{\alpha}, P_S) = \frac{I_{\mathrm{SKL}}(W; S)}{\alpha}.$$

*Proof.* Considering Lemma 2 and Lemma 3, we just need to verify that $J_E(w, s) = -\log P_{W|S}(w|s)$ can be decomposed into $J_E(w, s) = \frac{\alpha}{n} \sum_{i=1}^{n} \ell(w, z_i) + g(w) + h(s)$, for $\alpha > 0$. Note that

$$J_E(w, s) = -\log P_{W|S}^{\alpha}(w|s) = \alpha L_E(w, s) - \log \pi(w) + \log V(s, \alpha), \tag{34}$$

then we have:

$$I_{\text{SKL}}(W; S) = \mathbb{E}_{P_{W,S}}[J_P(W, P_S) - J_E(W, S)] \tag{35}$$
$$= \alpha \cdot \overline{\text{gen}}(P_{W|S}^{\alpha}, P_S). \qquad \square$$

Using Theorem 1, we can also derive the following lower bound on the expected generalization error in terms of total variation distance. As a comparison, an *upper* bound on the generalization error of a learning algorithm in terms of total variation distance is provided in [52].

**Corollary 2.** *For $(\alpha, \pi(w), L_E(w, s))$-Gibbs algorithm, the following lower bound on the generalization error of the Gibbs algorithm holds:*

$$\overline{\text{gen}}(P_{W|S}^{\alpha}, P_S) \geq \frac{TV^2(P_{W,S}, P_W \otimes P_S)}{\alpha}, \tag{36}$$

*where*

$$TV(P_{W,S}, P_W \otimes P_S) \triangleq \int \int \left| P_{W,S}(w, s) - P_W(w) P_S(s) \right| dw ds \tag{37}$$

*denotes total variation distance.*

*Proof.* This can be proved immediately by combining Theorem 1 with the well-known Pinsker's inequality [49],

$$TV(P_{W,S}, P_W \otimes P_S) \leq \sqrt{2 \min(I(W; S), L(W; S))}. \tag{38}$$

Note that the lower bound in Corollary 2 is bounded in $[0, \frac{4}{\alpha}]$. $\qquad \square$

## B.2 General Properties

In this section, we provide more discussions about other properties of the symmetrized KL divergence, including data processing inequality, variational representation, chain rule, and their implications in learning problems.

**Data Processing Inequality:** As shown in [59], symmetrized KL divergence is an $f$-divergence. Thus, the data processing inequality holds, i.e., for Markov chain $S \leftrightarrow W \leftrightarrow W'$,

$$I_{\text{SKL}}(S; W) \geq I_{\text{SKL}}(S; W'). \tag{39}$$

Using the data processing inequality for mutual information, [17, 71] show that pre/post-processing improves generalization, since these techniques give tighter mutual information-based generalization error bounds. However, our Theorem 1 only holds for Gibbs algorithm, which cannot characterize the generalization error for all conditional distributions $P_{W'|S}$ induced by the post-processing $P_{W'|W}$ in the Markov chain. Thus, it is hard to conclude that the pre/post-processing will reduce the exact generalization error for Gibbs algorithm by directly applying the data processing inequality.

**Variational Representation:** It is well-known that the mutual information has the following variational characterization

$$I(W; S) = \inf_{Q_W} D(P_{W|S} \| Q_W | P_S) = \inf_{Q_W, Q_S} D(P_{W,S} \| Q_W \otimes Q_S), \tag{40}$$

which implies that the product-of-marginal distribution minimizes the KL divergence for a given joint distribution. One may think that the counterpart for lautum information would be $\inf_{Q_W} D(P_S \otimes Q_W \| P_{W,S})$, but it is not true as shown in [49]. In general, the product-of-marginal distribution does not minimize $D(Q_W \otimes Q_S \| P_{W,S})$, and lautum information satisfies the following variational characterization

$$L(W; S) = \inf_{Q_S} D(P_W \otimes P_S \| P_{W|S} \otimes Q_S). \tag{41}$$

Thus, the product-of-marginal distribution $P_S \otimes P_W$ does not minimize the symmetrized KL divergence $D_{\text{SKL}}(P_{W,S} \| Q_W \otimes Q_S)$.

**Chain Rule:** As shown in [17], using the chain rule of mutual information, i.e., $I(W; S) = \sum_{i=1}^{n} I(W; Z_i | Z^{i-1})$ and the fact that $I(W; Z_i | Z^{i-1}) \geq I(W; Z_i)$ for i.i.d. samples, the mutual information based generalization bound can be tightened by considering the individual sample mutual information $I(W; Z_i)$.

However, lautum information does not satisfy the same chain rule as mutual information in general, and it is hard to characterize the generalization error of Gibbs algorithm using individual terms $I_{\text{SKL}}(W; Z_i)$. To see this, we have the following example to show that the joint symmetrized KL information $I_{\text{SKL}}(W; S)$ can be either larger or smaller than the sum of individual terms $I_{\text{SKL}}(W; Z_i)$.

**Example 1.** *Consider the following joint distribution for binary random variables* $W, Z_1, Z_2 \in \{0, 1\}$,

$$P_{W,Z_1,Z_2}(w, z_1, z_2) = \begin{cases} \frac{1}{8}, & \text{if } (z_1, z_2) = (0, 0), \\ \frac{1}{4} - \epsilon, & \text{if } w = 1, \text{ and } (z_1, z_2) \neq (0, 0), \\ \epsilon, & \text{otherwise.} \end{cases} \tag{42}$$

*It can be verified that* $Z_1$ *and* $Z_2$ *are mutually independent Bernoulli random variable with* $p = \frac{1}{2}$*, and the conditional distribution is symmetric in the sense that* $P_{W|Z_1,Z_2}(w|0, 1) = P_{W|Z_1,Z_2}(w|1, 0)$*.*

**Case I:** *When* $\epsilon = 0.0001$*, we can compute the mutual information as*

$$I(W; Z_1) = I(W; Z_2) = 0.0943, \quad I(W; Z_1, Z_2) = 0.2014,$$

*which satisfies the bound* $I(W; Z_1, Z_2) \geq I(W; Z_1) + I(W; Z_2)$ *when* $Z_1 \perp Z_2$*. However, for lautum information*

$$L(W; Z_1) = L(W; Z_2) = 0.3257, \quad L(W; Z_1, Z_2) = 0.5315,$$

$L(W; Z_1) + L(W; Z_2) > L(W; Z_1, Z_2)$*, and*

$$I_{\text{SKL}}(W; Z_1) = I_{\text{SKL}}(W; Z_2) = 0.4200, \quad I_{\text{SKL}}(W; Z_1, Z_2) = 0.7329,$$
$$I_{\text{SKL}}(W; Z_1) + I_{\text{SKL}}(W; Z_2) > I_{\text{SKL}}(W; Z_1, Z_2).$$

**Case II:** *When* $\epsilon = 0.01$*, it can be verified that*

$$I_{\text{SKL}}(W; Z_1) = I_{\text{SKL}}(W; Z_2) = 0.1255, \quad I_{\text{SKL}}(W; Z_1, Z_2) = 0.2741,$$
$$I_{\text{SKL}}(W; Z_1) + I_{\text{SKL}}(W; Z_2) < I_{\text{SKL}}(W; Z_1, Z_2).$$

*Thus, individual sample symmetrized KL information cannot be used to characterize the behavior of* $I_{\text{SKL}}(W; S)$ *in general.*

## B.3 Example Details: Mean Estimation

### B.3.1 Generalization Error

We first evaluate the generalization error of the learning algorithm in (13) directly. Note that the output $W$ can be written as

$$W = \frac{\sigma_1^2}{\sigma_0^2} \mu_0 + \frac{\sigma_1^2}{\sigma^2} \sum_{i=1}^{n} Z_i + N, \quad \text{with} \quad \sigma_1^2 = \frac{\sigma_0^2 \sigma^2}{n\sigma_0^2 + \sigma^2} \tag{43}$$

where $N \sim \mathcal{N}(0, \sigma_1^2 I_d)$ is independent from the training samples $S = \{Z_i\}_{i=1}^n$. Thus,

$$\overline{\text{gen}}(P_{W|S}, P_S)$$
$$= \mathbb{E}_{P_{W,S}}[L_P(W, \mu) - L_E(W, S)]$$
$$= \mathbb{E}_{P_{W,S}}\Big[\mathbb{E}_{P_{\tilde{Z}}}\big[\|W - \tilde{Z}\|_2^2\big] - \frac{1}{n}\sum_{i=1}^n \|W - Z_i\|_2^2\Big]$$
$$\overset{(a)}{=} \mathbb{E}_{P_{W,Z_i} \otimes P_{\tilde{Z}}}\Big[(2W - \tilde{Z} - Z_i)^\top (Z_i - \tilde{Z})\Big]$$
$$= \mathbb{E}\Big[2\big(\frac{\sigma_1^2}{\sigma_0^2}\boldsymbol{\mu}_0 + \frac{\sigma_1^2}{\sigma^2}\sum_{i=1}^n Z_i + N\big)^\top (Z_i - \tilde{Z}) - (Z_i + \tilde{Z})^\top (Z_i - \tilde{Z})\Big]$$
$$\overset{(b)}{=} \frac{2\sigma_1^2}{\sigma^2}\mathbb{E}\Big[Z_i^\top (Z_i - \tilde{Z})\Big]$$
$$= \frac{2d\sigma_1^2 \sigma_Z^2}{\sigma^2} = \frac{2d\sigma_0^2 \sigma_Z^2}{n\sigma_0^2 + \sigma^2}, \tag{44}$$

where $\tilde{Z} \sim \mathcal{N}(\boldsymbol{\mu}, \sigma_Z^2 I_d)$ denotes an independent copy of the training sample, $(a)$ follows due to the fact that $Z^n$ are i.i.d, and $(b)$ follows from the fact that $Z_i - \tilde{Z}$ has zero mean, and it is only correlated with $Z_i$.

### B.3.2 Symmetrized KL Divergence

The following lemma from [49] characterizes the mutual and lautum information for the Gaussian channel.

**Lemma 4.** *[49, Theorem 14] Consider the following model*

$$\boldsymbol{Y} = \boldsymbol{A}\boldsymbol{X} + \boldsymbol{N}_{\text{G}}, \tag{45}$$

*where $\boldsymbol{X} \in \mathbb{R}^{d_X}$ denotes the input random vector with zero mean (not necessarily Gaussian), $\boldsymbol{A} \in \mathbb{R}^{d_Y \times d_X}$ denotes the linear transformation undergone by the input, $\boldsymbol{Y} \in \mathbb{R}^{d_Y}$ is the output vector, and $\boldsymbol{N}_{\text{G}} \in \mathbb{R}^{d_Y}$ is a Gaussian noise vector independent of $\boldsymbol{X}$. The input and the noise covariance matrices are given by $\boldsymbol{\Sigma}$ and $\boldsymbol{\Sigma}_{N_{\text{G}}}$. Then, we have*

$$I(\boldsymbol{X}; \boldsymbol{Y}) = \frac{1}{2}\text{tr}\big(\boldsymbol{\Sigma}_{N_{\text{G}}}^{-1}\boldsymbol{A}\boldsymbol{\Sigma}\boldsymbol{A}^\top\big) - D\big(P_{\boldsymbol{Y}}\|P_{N_{\text{G}}}\big), \tag{46}$$

$$L(\boldsymbol{X}; \boldsymbol{Y}) = \frac{1}{2}\text{tr}\big(\boldsymbol{\Sigma}_{N_{\text{G}}}^{-1}\boldsymbol{A}\boldsymbol{\Sigma}\boldsymbol{A}^\top\big) + D\big(P_{\boldsymbol{Y}}\|P_{N_{\text{G}}}\big). \tag{47}$$

In our example, the output $W$ can be written as

$$W = \frac{\sigma_1^2}{\sigma_0^2}\boldsymbol{\mu}_0 + \frac{\sigma_1^2}{\sigma^2}\sum_{i=1}^n Z_i + N = \frac{\sigma_1^2}{\sigma^2}\sum_{i=1}^n (Z_i - \boldsymbol{\mu}) + \frac{\sigma_1^2}{\sigma_0^2}\boldsymbol{\mu}_0 + \frac{n\sigma_1^2}{\sigma^2}\boldsymbol{\mu} + N, \tag{48}$$

where $N \sim \mathcal{N}(0, \sigma_1^2 I_d)$. Setting $P_{N_{\text{G}}} \sim \mathcal{N}(\frac{\sigma_1^2}{\sigma_0^2}\boldsymbol{\mu}_0 + \frac{n\sigma_1^2}{\sigma^2}\boldsymbol{\mu}, \sigma_1^2 I_d)$ and $\boldsymbol{\Sigma} = \sigma_Z^2 I_{nd}$ in Lemma 4 gives

$$\text{tr}\big(\boldsymbol{\Sigma}_{N_{\text{G}}}^{-1}\boldsymbol{A}\boldsymbol{\Sigma}\boldsymbol{A}^\top\big) = \text{tr}\big(\frac{\sigma_Z^2}{\sigma_1^2}\boldsymbol{A}\boldsymbol{A}^\top\big), \tag{49}$$

and noticing that $\boldsymbol{A}\boldsymbol{A}^\top = \frac{n\sigma_1^4}{\sigma^4}I_d$ completes the proof.

### B.4 ISMI Bound

In this subsection, we evaluate the following individual sample mutual information (ISMI) bound from [19, Theorem 2] for the example discussed in Section 2.2 with i.i.d. samples generated from Gaussian distribution $P_Z \sim \mathcal{N}(\boldsymbol{\mu}, \sigma_Z^2 I_d)$.

**Lemma 5.** *[19, Theorem 2]* Suppose $\ell(\widetilde{W}, \widetilde{Z})$ satisfies $\Lambda_{\ell(\widetilde{W},\widetilde{Z})}(\lambda) \leq \psi_+(\lambda)$ for $\lambda \in [0, b_+)$, and $\Lambda_{\ell(\widetilde{W},\widetilde{Z})}(\lambda) \leq \psi_-(-\lambda)$ for $\lambda \in (b_-, 0]$ under $P_{\widetilde{Z},\widetilde{W}} = P_Z \otimes P_W$, where $0 < b_+ \leq \infty$ and $-\infty \leq b_- < 0$. Then,

$$\text{gen}(P_{W|S}, P_S) \leq \frac{1}{n} \sum_{i=1}^{n} \psi_-^{*-1}\big(I(W; Z_i)\big), \tag{50}$$

$$-\text{gen}(P_{W|S}, P_S) \leq \frac{1}{n} \sum_{i=1}^{n} \psi_+^{*-1}\big(I(W; Z_i)\big). \tag{51}$$

We need to compute the mutual information between each individual sample and the output hypothesis $I(W; Z_i)$, and the CGF of $\ell(\widetilde{W}, \widetilde{Z})$, where $\widetilde{W}, \widetilde{Z}$ are independent copies of $W$ and $Z$ with the same marginal distribution, respectively.

Since $W$ and $Z_i$ are Gaussian, $I(W; Z_i)$ can be computed exactly using covariance matrix:

$$\text{Cov}[Z_i, W] = \begin{pmatrix} \sigma_Z^2 I_d & \frac{\sigma_1^2}{\sigma^2} \sigma_Z^2 I_d \\ \frac{\sigma_1^2}{\sigma^2} \sigma_Z^2 I_d & \big(\frac{n\sigma_1^4}{\sigma^4} \sigma_Z^2 + \sigma_1^2\big) I_d \end{pmatrix}, \tag{52}$$

then, we have

$$\begin{aligned} I(W; Z_i) &= \frac{d}{2} \log \frac{\frac{n\sigma_1^4}{\sigma^4} \sigma_Z^2 + \sigma_1^2}{\frac{(n-1)\sigma_1^4}{\sigma^4} \sigma_Z^2 + \sigma_1^2} \\ &= \frac{d}{2} \log \left(1 + \frac{\sigma_1^2 \sigma_Z^2}{(n-1)\sigma_1^2 \sigma_Z^2 + \sigma^4}\right) \\ &= \frac{d}{2} \log \left(1 + \frac{\sigma_0^2 \sigma_Z^2}{(n-1)\sigma_0^2 \sigma_Z^2 + n\sigma_0^2 \sigma^2 + \sigma^4}\right), \end{aligned} \tag{53}$$

for $i = 1, \cdots, n$, $n \geq 2$. In addition, since

$$W \sim \mathcal{N}\Big(\frac{\sigma_1^2}{\sigma_0^2}\boldsymbol{\mu}_0 + \frac{n\sigma_1^2}{\sigma^2}\boldsymbol{\mu}, \big(\frac{n\sigma_1^4}{\sigma^4}\sigma_Z^2 + \sigma_1^2\big)I_d\Big), \tag{54}$$

it can be shown that $\ell(\widetilde{W}, \widetilde{Z}) = \|\widetilde{Z} - \widetilde{W}\|^2$ is a scaled non-central chi-square distribution with $d$ degrees of freedom, where the scaling factor $\sigma_\ell^2 \triangleq (\frac{n\sigma_1^4}{\sigma^4} + 1)\sigma_Z^2 + \sigma_1^2$ and its non-centrality parameter $\eta \triangleq \frac{\sigma^2}{n\sigma_0^2 + \sigma^2}\|\boldsymbol{\mu}_0 - \boldsymbol{\mu}\|_2^2$.

Note that the expectation of chi-square distribution with non-centrality parameter $\eta$ and $d$ degrees of freedom is $d + \eta$ and its moment generating function is $\exp(\frac{\eta\lambda}{1-2\lambda})(1 - 2\lambda)^{-d/2}$. Therefore, the CGF of $\ell(\widetilde{W}, \widetilde{Z})$ is given by

$$\Lambda_{\ell(\widetilde{W},\widetilde{Z})}(\lambda) = -(d\sigma_\ell^2 + \eta)\lambda + \frac{\eta\lambda}{1 - 2\sigma_\ell^2\lambda} - \frac{d}{2}\log(1 - 2\sigma_\ell^2\lambda), \tag{55}$$

for $\lambda \in (-\infty, \frac{1}{2\sigma_\ell^2})$. Since $\text{gen}(P_{W|S}, P_Z) \geq 0$, we only need to consider the case $\lambda < 0$. It can be shown that:

$$\begin{aligned} \Lambda_{\ell(\widetilde{W},\widetilde{Z})}(\lambda) &= -d\sigma_\ell^2\lambda - \frac{d}{2}\log(1 - 2\sigma_\ell^2\lambda) + \frac{2\sigma_\ell^2\eta\lambda^2}{1 - 2\sigma_\ell^2\lambda} \\ &= \frac{d}{2}(-u - \log(1 - u)) + \frac{2\sigma_\ell^2\eta\lambda^2}{1 - 2\sigma_\ell^2\lambda}, \end{aligned} \tag{56}$$

where $u \triangleq 2\sigma_\ell^2\lambda$. Further note that

$$-u - \log(1 - u) \leq \frac{u^2}{2}, \ u < 0, \tag{57}$$

$$\frac{2\sigma_\ell^2\eta\lambda^2}{1 - 2\sigma_\ell^2\lambda} \leq 2\sigma_\ell^2\eta\lambda^2, \ \lambda < 0. \tag{58}$$

We have the following upper bound on the CGF of $\ell(\widetilde{W}, \widetilde{Z})$:

$$\Lambda_{\ell(\widetilde{W},\widetilde{Z})}(\lambda) \leq (d\sigma_\ell^4 + 2\sigma_\ell^2\eta)\lambda^2, \quad \lambda < 0, \tag{59}$$

which means that $\ell(\widetilde{W}, \widetilde{Z})$ is $\sqrt{d\sigma_\ell^4 + 2\sigma_\ell^2\eta}$-sub-Gaussian for $\lambda < 0$. Combining the results in (53), Lemma 5 gives the following bound

$$\overline{\mathrm{gen}}(P_{W|S}, P_S) \leq \sqrt{\frac{d^2\sigma_\ell^4 + 2d\sigma_\ell^2\eta}{2} \log(1 + \frac{\sigma_0^2\sigma_Z^2}{(n-1)\sigma_0^2\sigma_Z^2 + n\sigma_0^2\sigma^2 + \sigma^4})}. \tag{60}$$

If $\sigma^2 = \frac{n}{2\alpha}$ is a constant, i.e., $\alpha = \mathcal{O}(n)$, then as $n \to \infty$, $\sigma_1^2 = \mathcal{O}(\frac{1}{n})$ and $\sigma_\ell^2 = \mathcal{O}(1)$, and the above bound is $\mathcal{O}\left(\frac{1}{\sqrt{n}}\right)$.

# C  Expected Generalization Error Upper Bound

## C.1  Proof of Theorem 2

We prove a slightly more general form of Theorem 2 as follows:

**Theorem 4.** *Suppose that the training samples $S = \{Z_i\}_{i=1}^n$ are i.i.d generated from the distribution $P_Z$ and the loss function $\ell(w, Z)$ satisfies $\Lambda_{\ell(w,Z)}(\lambda) \leq \psi(-\lambda)$, for $\lambda \in (-b, 0)$ and $0 < b$ under data-generating distribution $P_Z$ for all $w \in \mathcal{W}$. Let us assume $\exists C_E \in \mathbb{R}_0^+$ such that $\frac{L(W;S)}{I(W;S)} \geq C_E$, and we further assume:*

$$\exists\, 0 < \kappa < \infty, \quad s.t. \quad \psi^{\star-1}(\frac{\kappa}{n}) - \frac{(1 + C_E)\kappa}{\alpha} = 0. \tag{61}$$

*Then, the following upper bound holds for the expected generalization error of $(\alpha, \pi(w), L_E(w, s))$-Gibbs algorithm:*

$$0 \leq \overline{\mathrm{gen}}(P_{W|S}^\alpha, P_S) \leq \frac{(1 + C_E)\kappa}{\alpha}. \tag{62}$$

*Proof.* It is shown in [19, Proposition 2] that the following generalization error bound holds,

$$\overline{\mathrm{gen}}(P_{W|S}^\alpha, P_S) \leq \psi^{\star-1}\left(\frac{I(W;S)}{n}\right). \tag{63}$$

By Theorem 1 and the assumption on $C_E$, we have

$$\overline{\mathrm{gen}}(P_{W|S}^\alpha, P_S) = \frac{I(W;S) + L(W;S)}{\alpha} \geq \frac{(1 + C_E)I(W;S)}{\alpha}. \tag{64}$$

Therefore,

$$\frac{(1 + C_E)I(W;S)}{\alpha} \leq \psi^{\star-1}\left(\frac{I(W;S)}{n}\right). \tag{65}$$

Consider the function $F(u) \triangleq \psi^{\star-1}(\frac{u}{n}) - \frac{(1+C_E)u}{\alpha}$, which is concave and satisfies $F(0) = 0$ by Lemma 1. If there exists $0 < \kappa < \infty$, such that $F(\kappa) = 0$, then $F(I(W;S)) \geq 0$ implies that

$$0 \leq I(W;S) \leq \kappa.$$

Since $\psi^{\star-1}(\cdot)$ is non-decreasing, we have

$$\overline{\mathrm{gen}}(P_{W|S}^\alpha, P_S) \leq \psi^{\star-1}\left(\frac{\kappa}{n}\right) = \frac{(1 + C_E)\kappa}{\alpha}. \qquad \square$$

In the following, we specify the different forms of $\psi(\lambda)$ function in Theorem 4 to capture different tail behaviors of the loss function. We first consider the $\sigma$-sub-Gaussian assumption.

**Theorem 2.** *(restated) Suppose that the training samples $S = \{Z_i\}_{i=1}^n$ are i.i.d generated from the distribution $P_Z$, and the non-negative loss function $\ell(w, Z)$ is $\sigma$-sub-Gaussian on the left-tail under distribution $P_Z$ for all $w \in \mathcal{W}$. We further assume $C_E \leq \frac{L(W;S)}{I(W;S)}$ for some $C_E \geq 0$. Then, for the $(\alpha, \pi(w), L_E(w, s))$-Gibbs algorithm, we have*

$$0 \leq \overline{\mathrm{gen}}(P_{W|S}^\alpha, P_S) \leq \frac{2\sigma^2\alpha}{(1 + C_E)n}.$$

*Proof.* If the loss function is $\sigma$-sub-Gaussian on the left-tail we have $\psi^{\star-1}(y) = \sqrt{2\sigma^2 y}$. Using Theorem 4 we have

$$\sqrt{2\sigma^2 \frac{\kappa}{n}} - \frac{(1+C_E)\kappa}{\alpha} = 0, \tag{66}$$

and the solution is $\kappa = \frac{2\sigma^2}{n} \frac{\alpha^2}{(1+C_E)^2}$. Therefore,

$$\overline{\text{gen}}(P_{W|S}^\alpha, P_S) \leq \frac{(1+C_E)\kappa}{\alpha} = \frac{2\sigma^2\alpha}{n(1+C_E)}. \qquad \square$$

### C.2 Other Tail Distributions

In this section, we consider the sub-Exponential and sub-Gamma assumptions for the loss function and it is shown that the rates of convergence in these two cases are the same as that of the sub-Gaussian assumption, i.e., $\mathcal{O}(1/n)$.

We first consider the sub-Exponential case.

**Corollary 3.** *Suppose that the training samples $S = \{Z_i\}_{i=1}^n$ are i.i.d generated from the distribution $P_Z$, and the non-negative loss function $\ell(w, Z)$ is $(\sigma_e^2, b)$-sub-Exponential on the left-tail under distribution $P_Z$ for all $w \in \mathcal{W}$. We further assume $C_E \leq \frac{L(W;S)}{I(W;S)}$ for some $C_E \geq 0$. Then, for the $(\alpha, \pi(w), L_E(w, s))$-Gibbs algorithm, we have*

$$\overline{\text{gen}}(P_{W|S}^\alpha, P_S) \leq \begin{cases} \frac{2\sigma_e^2\alpha}{n(1+C_E)}, & \text{if } n \geq \frac{2bI(W;S)}{\sigma_e^2}; \\ \frac{\sigma_e^2}{2b}\left(\frac{\alpha b}{(n(1+C_E)-\alpha b)} + 1\right), & \text{if } \lceil\frac{\alpha b}{1+C_E}\rceil < n < \frac{2bI(W;S)}{\sigma_e^2}. \end{cases} \tag{67}$$

*Proof.* If the loss function is sub-Exponential on the left-tail we have

$$\psi^{\star-1}(y) = \begin{cases} \sqrt{2\sigma_e^2 y}, & \text{if } y \leq \frac{\sigma_e^2}{2b}; \\ by + \frac{\sigma_e^2}{2b}, & \text{otherwise.} \end{cases}$$

If $\frac{I(W;S)}{n} \leq \frac{\sigma_e^2}{2b}$, by Theorem 4, we have

$$\frac{(1+C_E)I(W;S)}{\alpha} \leq \sqrt{2\sigma_e^2 \frac{I(W;S)}{n}}, \tag{68}$$

then the following upper bound holds,

$$I(W;S) \leq \frac{2\sigma_e^2\alpha^2}{(1+C_E)^2 n}, \tag{69}$$

which gives

$$\overline{\text{gen}}(P_{W|S}^\alpha, P_S) \leq \frac{2\sigma_e^2\alpha}{n(1+C_E)}. \tag{70}$$

If $\frac{I(W;S)}{n} > \frac{\sigma_e^2}{2b}$, we have

$$\frac{I(W;S)(1+C_E)}{\alpha} \leq \frac{bI(W;S)}{n} + \frac{\sigma_e^2}{2b}, \tag{71}$$

then the following upper bound holds when $n > \frac{\alpha b}{1+C_E}$,

$$I(W;S) \leq \frac{\alpha n \sigma_e^2}{2b(n(1+C_E)-\alpha b)}, \tag{72}$$

which gives

$$\overline{\text{gen}}(P_{W|S}^\alpha, P_S) \leq \frac{\sigma_e^2}{2b}\left(\frac{\alpha b}{(n(1+C_E)-\alpha b)} + 1\right). \qquad \square$$

Note that all the sub-Exponential loss functions are also sub-Exponential on the left-tail under the same distribution (the converse statement is not true).

The authors in [48, 58] also consider the sub-Exponential assumption for general learning algorithms and provide PAC-Bayesian upper bounds. The result in Corollary 3 is an upper bound on the expected generalization error for Gibbs algorithm under sub-Exponential assumption, which establishes the $\mathcal{O}(1/n)$ convergence rate.

Next, we provide an upper bound under sub-Gamma assumption.

**Corollary 4.** *Suppose that the training samples $S = \{Z_i\}_{i=1}^n$ are i.i.d generated from the distribution $P_Z$, and the non-negative loss function $\ell(w, Z)$ is $\Gamma(\sigma_s^2, c_s)$-sub-Gamma on the left-tail under distribution $P_Z$ for all $w \in \mathcal{W}$. We further assume $C_E \leq \frac{L(W;S)}{I(W;S)}$ for some $C_E \geq 0$. Then, for the $(\alpha, \pi(w), L_E(w, s))$-Gibbs algorithm, if $n > \frac{c_s\alpha}{(1+C_E)}$, we have*

$$\overline{gen}(P_{W|S}^\alpha, P_S) \leq \frac{2\sigma_s^2\alpha}{(1+C_E)n - \alpha c_s}\Big(1 + \frac{\alpha c_s}{(1+C_E)n - \alpha c_s}\Big). \tag{73}$$

*Proof.* By considering $\psi^{\star-1}(y) = \sqrt{2\sigma_s^2 y} + cy$ in Theorem 4, we have

$$\frac{(1+C_E)I(W;S)}{\alpha} \leq \sqrt{2\sigma_s^2\frac{I(W;S)}{n}} + c_s\frac{I(W;S)}{n}. \tag{74}$$

Then the following upper bound holds when $n > \frac{c_s\alpha}{(1+C_E)}$,

$$I(W;S) \leq \Big(\frac{\alpha}{(1+C_E)n - \alpha c_s}\Big)^2 2n\sigma_s^2, \tag{75}$$

which gives

$$\overline{gen}(P_{W|S}^\alpha, P_S) \leq \frac{2\sigma_s^2\alpha(1+C_E)n}{\big((1+C_E)n - \alpha c_s\big)^2}. \qquad \square \tag{76}$$

The sub-Gamma assumption is also considered in [1, 26] and PAC-Bayesian upper bounds are provided. Our Corollary 4 provides an upper bound on the expected generalization error for Gibbs algorithm under sub-Gamma assumption, which establishes the $\mathcal{O}(1/n)$ convergence rate.

# D  PAC-Bayesian Upper Bound

Since the $(\alpha, \pi(w), L_P(w, P_{S'}))$-Gibbs distribution only depends on the population risk $L_P(w, P_{S'})$ and is independent of the samples $S$, we can denote it as $P_W^{\alpha, L'_P}$. The following lemma provides an operational interpretation of the symmetrized KL divergence between the Gibbs posterior $P_{W|S}^\alpha$ and the prior distribution $P_W^{\alpha, L'_P}$.

**Lemma 6.** *Let us denote the $(\alpha, \pi(w), L_E(w, s))$-Gibbs algorithm as $P_{W|S}^\alpha$ and the $(\alpha, \pi(w), L_P(w, P_{S'}))$-Gibbs algorithm as $P_W^{\alpha, L'_P}$. Then, the following equality holds for these two Gibbs distributions with the same inverse temperature and prior distribution*

$$\mathbb{E}_{\Delta(P_{W|S=s}^\alpha, P_W^{\alpha, L'_P})}[L_P(W, P_{S'}) - L_E(W, s)] = \frac{D_{\mathrm{SKL}}(P_{W|S=s}^\alpha \| P_W^{\alpha, L'_P})}{\alpha}, \tag{76}$$

*where $\mathbb{E}_{\Delta(P_{W|S=s}^\alpha, P_W^{\alpha, L'_P})}[f(W)] = \mathbb{E}_{P_{W|S=s}^\alpha}[f(W)] - \mathbb{E}_{P_W^{\alpha, L'_P}}[f(W)]$.*

*Proof.*

$$D_{\text{SKL}}(P_{W|S=s}^{\alpha}\|P_W^{\alpha,L'_P})$$

$$= \mathbb{E}_{P_{W|S=s}^{\alpha}}\left[\log\frac{P_{W|S=s}^{\alpha}}{P_W^{\alpha,L'_P}}\right] - \mathbb{E}_{P_W^{\alpha,L'_P}}\left[\log\frac{P_{W|S=s}^{\alpha}}{P_W^{\alpha,L'_P}}\right]$$

$$\overset{(a)}{=} \mathbb{E}_{\Delta(P_{W|S=s}^{\alpha},P_W^{\alpha,L'_P})}\left[\log(e^{-\alpha(L_E(W,s)-L_P(W,P_{S'}))})\right]$$

$$= \alpha\,\mathbb{E}_{\Delta(P_{W|S=s}^{\alpha},P_W^{\alpha,L'_P})}\left[L_P(W,P_{S'}) - L_E(W,s)\right], \tag{77}$$

where (a) follows by the fact that partition functions $V(s,\alpha)$ do not depend on $W$. $\qquad\square$

**Theorem 3.** *(restated) Suppose that the training samples $S = \{Z_i\}_{i=1}^n$ are i.i.d generated from the distribution $P_Z$, and the non-negative loss function $\ell(w,Z)$ is $\sigma$-sub-Gaussian under data-generating distribution $P_Z$ for all $w \in \mathcal{W}$. If we use the $(\alpha,\pi(w),L_P(w,P_{S'}))$-Gibbs distribution as the PAC-Bayesian prior, where $P_{S'}$ is an arbitrary chosen (and known) distribution, the following upper bound holds for the generalization error of $(\alpha,\pi(w),L_E(w,s))$-Gibbs algorithm with probability at least $1 - 2\delta$, $0 < \delta < \frac{1}{2}$ under distribution $P_S$,*

$$\left|\mathbb{E}_{P_{W|S=s}^{\alpha}}[L_P(W,P_S) - L_E(W,s)]\right| \leq \frac{2\sigma^2\alpha}{(1+C_P(s))n} + \epsilon^2$$

$$+ 2\sqrt{\frac{\sigma^2\alpha}{(1+C_P(s))n}}\left(\sqrt[4]{2\sigma^2 D(P_{Z'}\|P_Z)} + \epsilon\right),$$

*where $\epsilon \triangleq \sqrt[4]{\frac{2\sigma^2\log(1/\delta)}{n}}$, and $C_P(s) \leq \frac{D\left(P_W^{\alpha,L'_P}\|P_{W|S=s}^{\alpha}\right)}{D\left(P_{W|S=s}^{\alpha}\|P_W^{\alpha,L'_P}\right)}$ for some $C_P(s) \geq 0$.*

*Proof.* Using Lemma 6, we have

$$D_{\text{SKL}}(P_{W|S}^{\alpha}\|P_W^{\alpha,L'_P}) = \alpha(\mathbb{E}_{P_{W|S=s}^{\alpha}}[L_P(W,P_{Z'})] - \mathbb{E}_{P_{W|S=s}^{\alpha}}[L_E(W,s)])$$

$$- \alpha(\mathbb{E}_{P_W^{\alpha,L'_P}}[L_P(W,P_{Z'})] - \mathbb{E}_{P_W^{\alpha,L'_P}}[L_E(W,s)])$$

$$\leq \alpha\left|\mathbb{E}_{P_{W|S=s}^{\alpha}}[L_P(W,P_{Z'})] - \mathbb{E}_{P_{W|S=s}^{\alpha}}[L_E(W,s)]\right|$$

$$+ \alpha\left|(\mathbb{E}_{P_W^{\alpha,L'_P}}[L_P(W,P_{Z'})] - \mathbb{E}_{P_W^{\alpha,L'_P}}[L_E(W,s)]\right|$$

$$\leq \alpha\left|\mathbb{E}_{P_{W|S=s}^{\alpha}}[L_P(W,P_{Z'})] - \mathbb{E}_{P_{W|S=s}^{\alpha}}[L_P(W,P_Z)]\right|$$

$$+ \alpha\left|\mathbb{E}_{P_{W|S=s}^{\alpha}}[L_P(W,P_Z)] - \mathbb{E}_{P_{W|S=s}^{\alpha}}[L_E(W,s)]\right|$$

$$+ \alpha\left|\mathbb{E}_{P_W^{\alpha,L'_P}}[L_P(W,P_{Z'})] - \mathbb{E}_{P_W^{\alpha,L'_P}}[L_P(W,P_Z)]\right|$$

$$+ \alpha\left|\mathbb{E}_{P_W^{\alpha,L'_P}}[L_P(W,P_Z)] - \mathbb{E}_{P_W^{\alpha,L'_P}}[L_E(W,s)]\right|, \tag{78}$$

and we just need to bound the four terms in the above inequality.

The first and the third term in (78) can be bounded using the Donsker-Varadhan variational characterization of KL divergence, note that for all $\lambda \in \mathbb{R}$,

$$D(P_{Z'}\|P_Z) \geq \mathbb{E}_{P_{Z'}}[\lambda\ell(w,Z')] - \log\mathbb{E}_{P_Z}[e^{\lambda\ell(w,Z)}]$$

$$\geq \lambda(L_P(w,P_{Z'}) - L_P(w,P_Z)) - \frac{\lambda^2\sigma^2}{2}, \tag{79}$$

where the last step follows from the sub-Gaussian assumption. Since the above inequality holds for all $\lambda \in \mathbb{R}$, the discriminant must be non-positive, which implies

$$|L_P(w,P_{Z'}) - L_P(w,P_Z)| \leq \sqrt{2\sigma^2 D(P_{Z'}\|P_Z)}, \quad \text{for all} \quad w \in \mathcal{W}. \tag{80}$$

We use the PAC-Bayesian bound in [29, Proposition 3] to bound the second and the fourth term in (78). For any posterior distribution $Q_{W|S=s}$, and prior distribution $Q_W$, if $\ell(w, Z)$ is $\sigma$-sub-Gaussian under $P_Z$ for all $w \in \mathcal{W}$, the following bound holds with probability $1 - \delta$,

$$\left| \mathbb{E}_{Q_{W|S=s}}[L_P(W, P_Z)] - \mathbb{E}_{Q_{W|S=s}}[L_E(W, s)] \right| \leq \sqrt{\frac{2\sigma^2 \big( D(Q_{W|S=s} \| Q_W) + \log(1/\delta) \big)}{n}}. \quad (81)$$

If we choose $P_{W|S}^\alpha$ as the posterior distribution and $P_W^{\alpha, L_P'}$ as the prior distribution, we have

$$\left| \mathbb{E}_{P_{W|S=s}^\alpha}[L_P(W, P_Z)] - \mathbb{E}_{P_{W|S=s}^\alpha}[L_E(W, s)] \right| \leq \sqrt{\frac{2\sigma^2 \left( D(P_{W|S=s}^\alpha \| P_W^{\alpha, L_P'}) + \log(1/\delta) \right)}{n}} \quad (82)$$

holds with probability $1 - \delta$. If we set $Q_{W|S=s} = Q_W = P_W^{\alpha, L_P'}$, we have

$$\left| \mathbb{E}_{P_W^{\alpha, L_P'}}[L_P(W, P_Z)] - \mathbb{E}_{P_W^{\alpha, L_P'}}[L_E(W, s)] \right| \leq \sqrt{\frac{2\sigma^2 \left( \log(1/\delta) \right)}{n}}. \quad (83)$$

Combining the bounds in (80), (82) and (83) with (78), we have

$$D_{\mathrm{SKL}}(P_{W|S}^\alpha \| P_W^{\alpha, L_P'}) \leq \alpha \sqrt{\frac{2\sigma^2 \left( D(P_{W|S=s}^\alpha \| P_{W|S}^{\alpha, L_P'}) + \log(1/\delta) \right)}{n}} \quad (84)$$
$$+ \alpha \sqrt{\frac{2\sigma^2 \left( \log(1/\delta) \right)}{n}} + 2\alpha \sqrt{2\sigma^2 D(P_{Z'} \| P_Z)}.$$

Then, using the assumption that $(1 + C_P(s)) D(P_{W|S=s}^\alpha \| P_W^{\alpha, L_P'}) \leq D_{\mathrm{SKL}}(P_{W|S}^\alpha \| P_{W|S}^{\alpha, L_P})$, we have

$$(1 + C_P(s)) D(P_{W|S=s}^\alpha \| P_W^{\alpha, L_P'}) \leq \alpha \sqrt{\frac{2\sigma^2 \left( D(P_{W|S=s}^\alpha \| P_W^{\alpha, L_P'}) + \log(1/\delta) \right)}{n}}$$
$$+ \alpha \sqrt{\frac{2\sigma^2 \left( \log(1/\delta) \right)}{n}} + 2\alpha \sqrt{2\sigma^2 D(P_{Z'} \| P_Z)}. \quad (85)$$

Denote $\alpha' \triangleq \frac{\alpha}{(1 + C_P(s))}$, then we have

$$D(P_{W|S=s}^\alpha \| P_W^{\alpha, L_P'}) - \sqrt{\frac{2\alpha'^2 \sigma^2 \left( \log(1/\delta) \right)}{n}} - \sqrt{8\alpha'^2 \sigma^2 D(P_{Z'} \| P_Z)}$$
$$\leq \sqrt{\frac{2\alpha'^2 \sigma^2 \left( D(P_{W|S=s}^\alpha \| P_W^{\alpha, L_P'}) + \log(1/\delta) \right)}{n}}. \quad (86)$$

If we have $0 \leq D(P_{W|S=s}^\alpha \| P_W^{\alpha, L_P'}) \leq \sqrt{\frac{2\alpha'^2 \sigma^2 (\log(1/\delta))}{n}} + \sqrt{8\alpha'^2 \sigma^2 D(P_{Z'} \| P_Z)}$, then the above inequality holds. Otherwise, we could take square over both sides in (86), and denote

$$A \triangleq C + \sqrt{\frac{2\sigma^2 \alpha'^2 \log(1/\delta)}{n}}, \quad B \triangleq \sqrt{8\alpha'^2 \sigma^2 D(P_{Z'} \| P_Z)},$$

where $C \triangleq \frac{\sigma^2 \alpha'^2}{n}$, then we have

$$D^2(P_{W|S=s}^\alpha \| P_W^{\alpha, L_P'}) - 2D(P_{W|S=s}^\alpha \| P_W^{\alpha, L_P'})(A + B) + B^2 + 2(A - C)B \leq 0. \quad (87)$$

Solving the above inequality gives:

$$0 \leq D(P_{W|S=s}^\alpha \| P_W^{\alpha, L_P'}) \leq \sqrt{A^2 + 2BC} + A + B. \quad (88)$$

As $\sqrt{x+y} \leq \sqrt{x} + \sqrt{y}$ for positive $x, y$ and $A \geq C$, we have

$$D(P_{W|S=s}^{\alpha} \| P_W^{\alpha, L_P'}) \leq 2A + B + \sqrt{2BC} \leq 2A + B + \sqrt{2AB} \leq (\sqrt{2A} + \sqrt{B})^2. \quad (89)$$

Now using (89) in (82) and applying the inequality $\sqrt{x+y} \leq \sqrt{x} + \sqrt{y}$, we have:

$$\left| \mathbb{E}_{P_{W|S=s}^{\alpha}} [L_P(W, \mu) - L_E(W, s)] \right|$$

$$\leq \sqrt{\frac{2\sigma^2(\sqrt{2A} + \sqrt{B})^2 + 2\sigma^2 \log(1/\delta)}{n}}$$

$$\leq \sqrt{\frac{4\sigma^2 A}{n}} + \sqrt{\frac{2\sigma^2 B}{n}} + \sqrt{\frac{2\sigma^2 \log(1/\delta)}{n}}$$

$$\leq \frac{2\alpha\sigma^2}{(1 + C_P(s))n} + \sqrt{\frac{2\sigma^2 (\log(1/\delta))}{n}}$$

$$+ 2\sqrt{\frac{\alpha\sigma^2}{(1 + C_P(s))n}} \left( \sqrt[4]{\frac{2\sigma^2 \log(1/\delta)}{n}} + \sqrt[4]{2\sigma^2 D(P_{Z'} \| P_Z)} \right).$$

As both (82) and (83) hold with probability at least $1 - \delta$, the above inequality holds with probability at least $1 - 2\delta$ by the union bound [67]. □

## E  Asymptotic Behavior of Generalization Error for Gibbs Algorithm

### E.1  Large Inverse Temperature Details

**Proposition 1.** *(restated)* *In the single-well case, if the Hessian matrix $H^*(S)$ is not singular, then the generalization error of the $(\infty, \pi(\boldsymbol{w}), L_E(\boldsymbol{w}, s))$-Gibbs algorithm is*

$$\overline{gen}(P_{W|S}^{\infty}, P_S) = \mathbb{E}_{\Delta_{W,S}} \left[ \frac{1}{2} W^\top H^*(S) W \right]$$

$$+ \mathbb{E}_{P_S} \left[ (W^*(S) - \mathbb{E}[W^*(S)])^\top (H^*(S)W^*(S) - \mathbb{E}[H^*(S)W^*(S)]) \right],$$

*where* $\mathbb{E}_{\Delta_{W,S}}[f(W, S)] \triangleq \mathbb{E}_{P_W \otimes P_S}[f(W, S)] - \mathbb{E}_{P_{W,S}}[f(W, S)]$.

*Proof.* It is shown in [12, 33] that if the following Hessian matrix

$$H^*(S) = \nabla_w^2 L_E(w, S) \big|_{w=W^*(S)} \quad (90)$$

is not singular, then as $\alpha \to \infty$

$$P_{W|S}^{\alpha} \to \mathcal{N}(W^*(S), \frac{1}{\alpha} H^*(S)^{-1}) \quad (91)$$

in distribution. Then, the mean of the marginal distribution $P_W$ equals to the mean of $W^*(S)$, i.e.,

$$\mathbb{E}_{P_W}[W] = \mathbb{E}_{P_S}[W^*(S)]. \quad (92)$$

To apply Theorem 1, we evaluate the symmetrized KL information using the Gaussian approximation:

$$I(W; S) + L(W; S)$$

$$= \mathbb{E}_{P_{W,S}}[\log P_{W|S}^{\alpha}] - \mathbb{E}_{P_W \otimes P_S}[\log P_{W|S}^{\alpha}]$$

$$= \mathbb{E}_{P_{W,S}} \left[ -\frac{\alpha}{2}(W - W^*(S))^\top H^*(S)(W - W^*(S)) \right]$$

$$+ \mathbb{E}_{P_W \otimes P_S} \left[ \frac{\alpha}{2}(W - W^*(S))^\top H^*(S)(W - W^*(S)) \right]$$

$$= \mathbb{E}_{P_W \otimes P_S} \left[ \frac{\alpha}{2} W^\top H^*(S) W \right] - \mathbb{E}_{P_{W,S}} \left[ \frac{\alpha}{2} W^\top H^*(S) W \right]$$

$$+ \mathbb{E}_{P_S \otimes P_W} \left[ \frac{\alpha}{2} \left( \text{tr}\big(H^*(S)(W^*(S)W^*(S)^\top - WW^*(S)^\top - W^*(S)W^\top)\big) \right) \right]$$

$$- \mathbb{E}_{P_S \otimes P_{W|S}} \left[ \frac{\alpha}{2} \left( \text{tr}\big(H^*(S)(W^*(S)W^*(S)^\top - WW^*(S)^\top - W^*(S)W^\top)\big) \right) \right]. \quad (93)$$

Note that $\mathbb{E}_{P_W}[W] = \mathbb{E}_{P_S}[W^*(S)]$ and $\mathbb{E}_{P_{W|S}}[W] = W^*(S)$, we have

$$
\begin{aligned}
\overline{\text{gen}}(P_{W|S}^\infty, \mu) &= \frac{I(W;S) + L(W;S)}{\alpha} \\
&= \mathbb{E}_{P_W \otimes P_S}\Big[\frac{1}{2}W^\top H^*(S)W\Big] - \mathbb{E}_{P_{W,S}}\Big[\frac{1}{2}W^\top H^*(S)W\Big] \\
&\quad + \mathbb{E}_{P_S}\Big[\frac{1}{2}\Big(\text{tr}\big(H^*(S)(-\mathbb{E}[W^*(S)]W^*(S)^\top - W^*(S)\mathbb{E}[W^*(S)]^\top)\big)\Big)\Big] \\
&\quad - \mathbb{E}_{P_S}\Big[\frac{1}{2}\Big(\text{tr}\big(H^*(S)(-W^*(S)W^*(S)^\top - W^*(S)W^*(S)^\top)\big)\Big)\Big] \\
&= \mathbb{E}_{P_W \otimes P_S}\Big[\frac{1}{2}W^\top H^*(S)W\Big] - \mathbb{E}_{P_{W,S}}\Big[\frac{1}{2}W^\top H^*(S)W\Big] \\
&\quad + \mathbb{E}_{P_S}\Big[(W^*(S) - \mathbb{E}[W^*(S)])^\top (H^*(S)W^*(S) - \mathbb{E}[H^*(S)W^*(S)])\Big]. \quad \square
\end{aligned}
$$

**Proposition 2.** *(restated) If we assume that $\pi(W)$ is a uniform distribution over $\mathcal{W}$, and the Hessian matrices $H_u^*(S)$ are not singular for all $u \in \{1, \cdots, M\}$, then the generalization error of the $(\infty, \pi(\boldsymbol{w}), L_E(\boldsymbol{w}, s))$-Gibbs algorithm in the multiple-well case can be bounded as*

$$
\begin{aligned}
\overline{\text{gen}}(P_{W|S}^\infty, P_S) &\le \frac{1}{M}\sum_{u=1}^M \Big[\mathbb{E}_{\Delta_{W_u, S}}\Big[\frac{1}{2}W_u^\top H_u^*(S)W_u\Big] \\
&\quad + \mathbb{E}_{P_S}\Big[(W_u^*(S) - \mathbb{E}[W_u^*(S)])^\top H_u(W_u^*(S) - \mathbb{E}[W_u^*(S)])\Big]\Big].
\end{aligned}
$$

*Proof.* In this multiple-well case, it is shown in [12] that the Gibbs algorithm can be approximated by the following Gaussian mixture distribution

$$
P_{W|S}^\alpha \to \frac{1}{\sum_{u=1}^M \pi(W_u^*(S))}\sum_{u=1}^M \pi\big(W_u^*(S)\big)\mathcal{N}\big(W_u^*(S), \frac{1}{\alpha}H_u^*(S)^{-1}\big), \tag{94}
$$

as long as $H_u^*(S) \triangleq \nabla_w^2 L_E(w, S)\big|_{w=W_u^*(S)}$ is not singular for all $u \in \{1, \cdots, M\}$.

However, there is no closed form for the symmetrized KL information for Gaussian mixtures. Thus, we use Theorem 1 to construct an upper bound of the generalization error.

Consider the latent random variable $U \in \{1, \cdots, M\}$ which denotes the index of the Gaussian component of $P_{W|S}^\alpha$. Then, conditioning on $U$ and $S$, $W$ is a Gaussian random variable. Moreover, since $\pi(W)$ is a uniform prior, $U$ is a discrete uniform distribution $P_U(U = u) = \frac{1}{M}$, and $U \perp S$. Note that for mutual information, we have

$$
I(S;W|U) = I(S;W|U) + I(S;U) = I(S;W,U) = I(S;W) + I(S;U|W) \ge I(S;W), \tag{95}
$$

and for lautum information

$$
L(W;S) \overset{(a)}{\le} L(W,U;S) \overset{(b)}{=} L(U;S) + L(W;S|U) = L(W;S|U), \tag{96}
$$

where $(a)$ is due to the data processing inequality for any $f$-divergence, and $(b)$ follows by the fact that the chain rule of lautum information holds when $U \perp S$ as shown in [49].

Thus, we can upper bound $I(S;W)$ and $L(S;W)$ with $I(S;W|U)$ and $L(S;W|U)$, respectively,

$$
\begin{aligned}
&\overline{\text{gen}}(P_{W|S}^\infty, \mu) \\
&= \lim_{\alpha \to \infty}\frac{I(S;W) + L(S;W)}{\alpha} \\
&\le \lim_{\alpha \to \infty}\frac{I(S;W|U) + L(S;W|U)}{\alpha} \\
&= \mathbb{E}_U\Big[\mathbb{E}_{P_{W|U} \otimes P_S}\Big[\frac{1}{2}W^\top H(w_u^*(S), S)W\Big]\Big] - \mathbb{E}_U\Big[\mathbb{E}_{P_{W,S|U}}\Big[\frac{1}{2}W^\top H(w_U^*(S), S)W\Big]\Big] \\
&\quad + \mathbb{E}_U\Big[\mathbb{E}_{P_S}\Big[(w_U^*(S) - \mathbb{E}[w_U^*(S)])^\top (H(w_U^*(S), S)w_U^*(S) - \mathbb{E}[H(w_U^*(S), S)w_U^*(S)])\Big]\Big]. \square
\end{aligned}
$$

## E.2 Regularity Conditions for MLE

In this section, we present the regularity conditions required by the asymptotic normality [64] of maximum likelihood estimates.

**Assumption 1.** *Regularity Conditions for MLE:*

1. $f(z|\boldsymbol{w}) \neq f(z|\boldsymbol{w}')$ *for* $\boldsymbol{w} \neq \boldsymbol{w}'$.

2. $\mathcal{W}$ *is an open subset of* $\mathbb{R}^d$.

3. *The function* $\log f(z|\boldsymbol{w})$ *is three times continuously differentiable with respect to* $\boldsymbol{w}$.

4. *There exist functions* $F_1(z) : \mathcal{Z} \to \mathbb{R}$, $F_2(z) : \mathcal{Z} \to \mathbb{R}$ *and* $M(z) : \mathcal{Z} \to \mathbb{R}$, *such that*

$$\mathbb{E}_{Z \sim f(z|\boldsymbol{w})}[M(Z)] < \infty,$$

*and the following inequalities hold for any* $\boldsymbol{w} \in \mathcal{W}$,

$$\left| \frac{\partial \log f(z|\boldsymbol{w})}{\partial w_i} \right| < F_1(z), \qquad \left| \frac{\partial^2 \log f(z|\boldsymbol{w})}{\partial w_i \partial w_j} \right| < F_1(z),$$

$$\left| \frac{\partial^3 \log f(z|\boldsymbol{w})}{\partial w_i \partial w_j \partial w_k} \right| < M(z), \qquad i, j, k = 1, 2, \cdots, d.$$

5. *The following inequality holds for an arbitrary* $\boldsymbol{w} \in \mathcal{W}$,

$$0 < \mathbb{E}_{Z \sim f(z|\boldsymbol{w})} \left[ \frac{\partial \log f(z|\boldsymbol{w})}{\partial w_i} \frac{\partial \log f(z|\boldsymbol{w})}{\partial w_j} \right] < \infty, \quad i, j = 1, 2, \cdots, d.$$

## E.3 Bayesian Learning Algorithm

In this section, we show that the symmetrized KL information can be used to characterize the generalization error of Gibbs algorithm in a different asymptotic regime, i.e., inverse temperature $\alpha = n$, then $\alpha$ and $n$ go to infinity simultaneously. In this regime, the Gibbs algorithm is equivalent to the Bayesian posterior distribution instead of ERM.

Suppose that we have $n$ i.i.d. training samples $S = \{Z_i\}_{i=1}^n$ generated from the distribution $P_Z$ defined on $\mathcal{Z}$, and we want to fit the training data with a parametric distribution family $\{f(z_i|\boldsymbol{w})\}_{i=1}^n$, where $\boldsymbol{w} \in \mathcal{W} \subset \mathbb{R}^d$ denotes the parameter and $\pi(\boldsymbol{w})$ denotes a pre-selected prior distribution. Here, the true data-generating distribution may not belong to the parametric family, i.e., $P_Z \neq f(\cdot|\boldsymbol{w})$ for $\boldsymbol{w} \in \mathcal{W}$. The following Bayesian posterior distribution

$$P_{W|S}(\boldsymbol{w}|z^n) = \frac{\pi(\boldsymbol{w}) \prod_i^n f(z_i|\boldsymbol{w})}{V(z^n)}, \quad \text{with} \quad V(z^n) = \int \pi(\boldsymbol{w}) \prod_i^n f(z_i|\boldsymbol{w}) dw, \qquad (97)$$

is equivalent to the $(n, \pi(\boldsymbol{w}), L_E(\boldsymbol{w}, s))$-Gibbs algorithm with log-loss $\ell(\boldsymbol{w}, z) = -\log f(z|\boldsymbol{w})$. Thus, Theorem 1 can be applied directly, and we just need to evaluate $I_{\text{SKL}}(W; S)$.

We further assume that the parametric family $\{f(z|\boldsymbol{w}), \boldsymbol{w} \in \mathcal{W}\}$ and prior $\pi(\boldsymbol{w})$ satisfy all the regularization conditions required for the Bernstein–von-Mises theorem [64] and the asymptotic Normality of the maximum likelihood estimate (MLE), including Assumption 1 and the condition that $\pi(w)$ is continuous and $\pi(w) > 0$ for all $w \in \mathcal{W}$.

In the asymptotic regime $n \to \infty$, Bernstein–von-Mises theorem under model mismatch [38, 64] states that we could approximate the Bayesian posterior distribution $P_{W|S}$ in (97) by

$$\mathcal{N}(\hat{W}_{\text{ML}}, \frac{1}{n} J(\boldsymbol{w}^*)^{-1}), \quad \text{where} \quad \hat{W}_{\text{ML}} \triangleq \underset{\boldsymbol{w} \in \mathcal{W}}{\arg\max} \sum_{i=1}^n \log f(Z_i|\boldsymbol{w}), \qquad (98)$$

denotes the MLE and

$$J(\boldsymbol{w}) \triangleq \mathbb{E}_Z\big[ -\nabla_{\boldsymbol{w}}^2 \log f(Z|\boldsymbol{w}) \big] \quad \text{with} \quad \boldsymbol{w}^* \triangleq \underset{\boldsymbol{w} \in \mathcal{W}}{\arg\min} D(P_Z \| f(\cdot|\boldsymbol{w})).$$

The asymptotic Normality of the MLE states that the distribution of $\hat{W}_{\mathrm{ML}}$ will converge to

$$\mathcal{N}\left(\boldsymbol{w}^*, \frac{1}{n}J(\boldsymbol{w}^*)^{-1}\mathcal{I}(\boldsymbol{w}^*)J(\boldsymbol{w}^*)^{-1}\right) \quad \text{with} \quad \mathcal{I}(\boldsymbol{w}) \triangleq \mathbb{E}_Z\left[\nabla_{\boldsymbol{w}}\log f(Z|\boldsymbol{w})\nabla_{\boldsymbol{w}}\log f(Z|\boldsymbol{w})^\top\right]$$

as $n \to \infty$. Thus, the marginal distribution $P_W$ can be approximated by a Gaussian distribution regardless the choice of prior $\pi(\boldsymbol{w})$.

Then, the symmetrized KL information can be computed using Lemma 4. By Theorem 1, we have

$$\overline{\text{gen}}(P_{W|S}, P_Z) = \frac{I_{\mathrm{SKL}}(S; W)}{n} = \frac{\mathrm{tr}(\mathcal{I}(\boldsymbol{w}^*)J(\boldsymbol{w}^*)^{-1})}{n}. \tag{99}$$

When the true model is in the parametric family $P_Z = f(\cdot|\boldsymbol{w}^*)$, we have $\mathcal{I}(\boldsymbol{w}^*) = J(\boldsymbol{w}^*)$, which gives the Fisher information matrix and $\overline{\text{gen}}(P_{W|S}, P_Z) = \frac{d}{n}$. This result suggests that the expected generalization error of MLE and that of the Bayesian posterior distribution are the same under suitable regularity conditions.

### E.4 Behavior of Empirical Risk

As an aside, we show that the empirical risk is a decreasing function of the inverse temperature $\alpha$. To see this, we first note that the derivative of $P_{W|S}^\alpha$ with respect to $\alpha$ is given by

$$\frac{\mathrm{d}P_{W|S}^\alpha(w|s)}{\mathrm{d}\alpha} = P_{W|S}^\alpha(w|s)\left(\mathbb{E}_{P_{W|S}^\alpha}[L_E(w, S)] - L_E(w, S)\right). \tag{100}$$

Then, we can compute the derivative of the empirical risk with respect to $\alpha$ as follows:

$$\begin{aligned}
\frac{\mathrm{d}\mathbb{E}_{P_{W,S}}[L_E(W, S)]}{\mathrm{d}\alpha} &= \mathbb{E}_{P_S}\left[\frac{\mathrm{d}\mathbb{E}_{P_{W|S}^\alpha}[L_E(W, S)]}{\mathrm{d}\alpha}\right] \\
&= \mathbb{E}_{P_S}\left[\int_{\mathcal{W}} L_E(w, S)\frac{\mathrm{d}P_{W|S}^\alpha(w|S)}{\mathrm{d}\alpha}dw\right] \\
&= \mathbb{E}_{P_S}\left[\int_{\mathcal{W}} P_{W|S}^\alpha(w|s)\left(L_E(w, S)\mathbb{E}_{P_{W|S}^\alpha}[L_E(w, S)] - L_E^2(w, S)\right)dw\right] \\
&= \mathbb{E}_{P_S}\left[\mathbb{E}_{P_{W|S}^\alpha}^2[L_E(w, S)] - \mathbb{E}_{P_{W|S}^\alpha}[L_E^2(w, S)]\right] \\
&= -\mathbb{E}_{P_S}[\mathrm{Var}_{P_{W|S}^\alpha}[L_E(W, S)]] \le 0
\end{aligned} \tag{101}$$

When $\alpha = 0$, it can be shown that $(0, \pi(w), L_E(w, s))$-Gibbs algorithm has zero generalization error. However, the empirical risk in this case could be large, since the training samples are not used at all. As $\alpha \to \infty$, the empirical risk is decreasing, but the generalization error could be large. Thus, the inverse temperature $\alpha$ controls the trade-off between the empirical risk and the generalization error.

## F  Regularized Gibbs Algorithm

### F.1  Proofs of Proposition 3 and Proposition 4

**Proposition 3.** *(restated) For $(\alpha, \pi(w), L_E(w, s) + \lambda R(w, s))$-Gibbs algorithm, its expected generalization error is given by*

$$\overline{\text{gen}}(P_{W|S}^\alpha, P_S) = \frac{I_{\mathrm{SKL}}(W; S)}{\alpha} - \lambda\mathbb{E}_{\Delta_{W,S}}[R(W, S)],$$

*where $\mathbb{E}_{\Delta_{W,S}}[R(W, S)] = \mathbb{E}_{P_W \otimes P_S}[R(W, S)] - \mathbb{E}_{P_{W,S}}[R(W, S)]$.*

*Proof.* For $(\alpha, \pi(w), L_E(w, s) + \lambda R(w, s))$-Gibbs algorithm, we have

$$\begin{aligned}
I_{\mathrm{SKL}}(W; S) &= \mathbb{E}_{P_{W,S}}[\log(P_{W|S}^\alpha)] - \mathbb{E}_{P_W \otimes P_S}[\log(P_{W|S}^\alpha)] \\
&= \alpha\left(\mathbb{E}_{P_W \otimes P_S}[L_E(W, S)] - \mathbb{E}_{P_{W,S}}[L_E(W, S)]\right) \\
&\quad + \alpha\lambda\left(\mathbb{E}_{P_W \otimes P_S}[R(W, S)] - \mathbb{E}_{P_{W,S}}[R(W, S)]\right) \\
&= \alpha\overline{\text{gen}}(P_{W|S}^\alpha, P_S) + \alpha\lambda\mathbb{E}_{\Delta_{W,S}}[R(W, S)]. \qquad \square
\end{aligned}$$

**Proposition 4.** *(restated) Suppose that we adopt the $\ell_2$-regularizer $R(w, s) = \|w - T(s)\|_2^2$, where $T(\cdot)$ is an arbitrary deterministic function $T : \mathcal{Z}^n \to \mathcal{W}$. Then, the expected generalization error of $(\alpha, \pi(w), L_E(w, s) + \lambda R(w, s))$-Gibbs algorithm is*

$$\overline{gen}(P_{W|S}^\alpha, P_S) = \frac{I_{\mathrm{SKL}}(W; S)}{\alpha} - \lambda\mathrm{tr}\big(\mathrm{Cov}[W, T(S)]\big),$$

*where $\mathrm{Cov}[W, T(S)]$ denotes the covariance matrix between $W$ and $T(S)$.*

*Proof.* We just need to compute $\mathbb{E}_{\Delta_{W,S}}[R(W, S)]$ by considering $R(w, s) = \|w - T(s)\|_2^2$,

$$\begin{aligned}
&\mathbb{E}_{P_W \otimes P_S}[R(W, S)] - \mathbb{E}_{P_{W,S}}[R(W, S)] \\
&= \mathbb{E}_{P_W \otimes P_S}\left[\|W - T(S)\|_2^2\right] - \mathbb{E}_{P_{W,S}}\left[\|W - T(S)\|_2^2\right] \\
&= \mathbb{E}_{P_{W,S}}\left[W^T T(S)\right] - \mathbb{E}_{P_W \otimes P_S}\left[W^T T(S)\right] \\
&= \mathrm{tr}(\mathrm{Cov}(W, T(S))). \hfill \square
\end{aligned}$$

### F.2 Generalization Error Upper Bounds for Regularized Gibbs Algorithm

For general regularization function $R(w, s)$, we can bound the $\mathbb{E}_{\Delta_{W,S}}[R(W, S)]$ term using the mutual information-based generalization error bound in [19, 71].

**Proposition 5.** *Suppose that the regularizer function $R(w, s)$ satisfies $\Lambda_{R(w,s)}(\lambda) \leq \psi(\lambda)$, for $\lambda \in (-b, b)$ and $b > 0$ under data-generating distribution $P_Z$ for all $w \in \mathcal{W}$. Then the following lower and upper bounds holds for $(\alpha, \pi(w), L_E(w, s) + \lambda R(w, s))$-Gibbs algorithm:*

$$\frac{I_{\mathrm{SKL}}(W; S)}{\alpha} - \lambda\psi^{*-1}(I(W; S)) \leq \overline{gen}(P_{W|S}^\alpha, P_S) \leq \frac{I_{\mathrm{SKL}}(W; S)}{\alpha} + \lambda\psi^{*-1}(I(W; S)) \quad (102)$$

*Proof.* Using the decoupling lemma from [19, Theorem 1], we have:

$$|\mathbb{E}_{\Delta_{W,S}}[R(W, S)]| \leq \psi^{*-1}(I(W; S)), \tag{103}$$

which means that

$$-\psi^{*-1}(I(W; S)) \leq \mathbb{E}_{\Delta_{W,S}}[R(W, S)] \leq \psi^{*-1}(I(W; S)). \tag{104}$$

The final results (102) follows directly from (104) and Proposition 3. $\hfill \square$

Note that the bounded CGF assumption is on the regularizer function $R(w, s)$. We could consider different assumptions on $\psi(\lambda)$ in Proposition 5 including sub-Gaussian, sub-Exponential and sub-Gamma. We focus on sub-Gaussian assumption for regularizer function in the following result.

**Corollary 5.** *Suppose that the regularizer function $R(w, s)$ is $\sigma$-sub-Gaussian under the distribution $P_S$ for all $w \in \mathcal{W}$. Then the following bounds holds for $(\alpha, \pi(w), L_E(w, s) + \lambda R(w, s))$-Gibbs algorithm:*

$$\frac{I_{\mathrm{SKL}}(W; S)}{\alpha} - \lambda\sqrt{2\sigma^2 I(W; S)} \leq \overline{gen}(P_{W|S}^\alpha, P_S) \leq \frac{I_{\mathrm{SKL}}(W; S)}{\alpha} + \lambda\sqrt{2\sigma^2 I(W; S)} \quad (105)$$

*Proof.* Considering $\psi^{*-1}(I(W; S)) = \sqrt{2\sigma^2 I(W; S)}$ in Proposition 5 completes the proof. $\hfill \square$

By assuming $\sigma$-sub-Gaussianity for both loss function and the regularizer, we provide a generalization error upper bound for regularized Gibbs algorithm in the following proposition.

**Proposition 6.** *Suppose that the training samples $S = \{Z_i\}_{i=1}^n$ are i.i.d generated from the distribution $P_Z$, and the non-negative loss function $\ell(w, Z)$ and the regularizer function $R(w, s)$ are $\sigma$-sub-Gaussian under data-generating distribution $P_Z$ for all $w \in \mathcal{W}$. We further assume $C_E = \frac{L(W;S)}{I(W;S)}$ for some $C_E \geq 0$. Then the following bounds holds for $(\alpha, \pi(w), L_E(w, s) + \lambda R(w, s))$-Gibbs algorithm:*

$$\overline{gen}(P_{W|S}^\alpha, P_S) \leq \begin{cases} \frac{2\sigma^2\alpha}{(1+C_E)}\left(\frac{1}{n} - \frac{\lambda}{\sqrt{n}}\right), & \text{if} \quad 0 \leq \lambda \leq \frac{1}{\sqrt{n}} \quad \text{and} \quad I(W;S) \leq \frac{2\sigma^2\alpha^2}{(1+C_E)^2}\left(\frac{1}{\sqrt{n}} - \lambda\right)^2; \\ \frac{2\sigma^2\alpha}{(1+C_E)}\left(\frac{1}{n} + \frac{\lambda}{\sqrt{n}}\right), & \text{otherwise.} \end{cases} \tag{106}$$

*Proof.* Using Proposition 5 and [71, Theorem 1], we have

$$\frac{I_{\mathrm{SKL}}(W;S)}{\alpha} - \lambda\sqrt{2\sigma^2 I(W;S)} \leq \min\left(\sqrt{\frac{2\sigma^2 I(W;S)}{n}}, \frac{I_{\mathrm{SKL}}(W;S)}{\alpha} + \lambda\sqrt{2\sigma^2 I(W;S)}\right).$$

If $\sqrt{\frac{2\sigma^2 I(W;S)}{n}} \leq \frac{I_{\mathrm{SKL}}(W;S)}{\alpha} + \lambda\sqrt{2\sigma^2 I(W;S)}$, and using $C_E I(W;S) = L(W;S)$, then we have:

$$\frac{I(W;S)(1 + C_E)}{\alpha} - \lambda\sqrt{2\sigma^2 I(W;S)} \leq \sqrt{\frac{2\sigma^2 I(W;S)}{n}}. \tag{107}$$

Solving (107) gives

$$I(W;S) \leq \frac{2\sigma^2 \alpha^2}{(1 + C_E)^2}\left(\frac{1}{\sqrt{n}} + \lambda\right)^2. \tag{108}$$

If $\frac{I_{\mathrm{SKL}}(W;S)}{\alpha} + \lambda\sqrt{2\sigma^2 I(W;S)} < \sqrt{\frac{2\sigma^2 I(W;S)}{n}}$, and using $C_E I(W;S) = L(W;S)$, then we have:

$$I(W;S) \leq \frac{2\sigma^2 \alpha^2}{(1 + C_E)^2}\left(\frac{1}{\sqrt{n}} - \lambda\right)^2, \tag{109}$$

for $0 \leq \lambda \leq \frac{1}{\sqrt{n}}$. Combining the (108) and (109) with [71, Theorem 1] completes the proof. $\qquad\square$

In Proposition 6, if $0 \leq \lambda \leq \frac{1}{\sqrt{n}}$ and $\frac{I(W;S)(1 + C_E)^2}{2\alpha^2\sigma^2} \leq \left(\frac{1}{\sqrt{n}} - \lambda\right)^2$ hold, then the upper bound would be tighter than the upper bound in Theorem 2 with $C_E = \frac{L(W;S)}{I(W;S)}$.