# OpenReview forum: "An Exact Characterization of the Generalization Error for the Gibbs Algorithm"
_NeurIPS.cc/2021/Conference — NeurIPS 2021 Poster_

### Official Review · Reviewer_9Z7C · 2021-07-14

**Rating:** 7
**Confidence:** 3

**Summary:**

This paper is concerned with characterizing (and more generally deriving upper bounds on)
the expected generalization error (as defined at (2)) for the Gibbs algorithm.
After giving some motivation for considering the Gibbs algorithm,
the authors state their key result: an exact characterization of the expected risk of the
algorithm, in terms of the symmetrized KL information between training sample and model,
and the temperature parameter of the Gibbs distribution.
This allows the authors to recover some known upper bounds under
iid sampling and bounded loss function, as well as deriving a new upper bound
when the loss function enjoys well-behaved tails
(sub-gaussian, sub-exponential, sub-gamma, but possibly unbounded).
This approach also yields a PAC-Bayes upper bound,
and an analysis of the behavior of the expected generalization error
when the temperature parameter shoots to infinity.

**Ethical Concerns:**

None.

**Limitations And Societal Impact:**

Adequately addressed.

**Main Review:**

The paper is well-written, the results improve in an elegant manner over the state-of-the-art in a well detailed manner, and the problem appears significant.  The results seem to be new, but it is possible that the reviewer is not aware of similar results having been published elsewhere. Good paper.

* This reviewer wonders whether the characterization in terms of symmetrized KL information is not well suited to also derive lower bounds.

* Minor comments
    * L.112 α-Réyni → α-Rényi
    * L.198 Gibbs algorithm → the Gibbs algorithm

**Time Spent Reviewing:**

3

---

> ### Author Response · Authors · 2021-08-09
> **Responding to Reviewers Comments**
>
> Thank you for taking the time to thoroughly read and comment on our paper.
>
> $\textbf{Reviewer's Comment:}$ This reviewer wonders whether the characterization in terms of symmetrized KL information is not well suited to also derive lower bounds.
>
> $\textbf{Response:}$ Your instinct is quite correct, and the paper does include such a lower bound, which we could better emphasize.  In particular, we derive a lower bound for the Gibbs algorithm in terms of total variation distance (see Corollary 2 in Appendix B.1) using the exact characterization of generalization error based on symmetrized KL information and Pinsker's inequality. As a comparison, generalization error upper bound based on total variation distance for all learning algorithms was studied in [49]. We will refine the exposition of the paper to better reveal this this point.
>
> Finally, all minor comments are addressed in the revised manuscript.

---

> > ### Comment · Reviewer_9Z7C · 2021-08-17
> > **Sore unchanged**
> >
> > Many thanks to the authors for their response.
> > This reviewer keeps the score unchanged: 7, Accept.

---

### Official Review · Reviewer_DrrA · 2021-07-16

**Rating:** 7
**Confidence:** 3

**Summary:**

This paper gives an exact characterization of the expected generalization error for the Gibbs algorithm by way of a symmetrized KL divergence between the input training data and the output hypothesis. This allows for tightening some existing generalization bounds under certain assumptions.


**Limitations And Societal Impact:**

Addressed by the authors. I do not foresee any potential negative societal impact of this work

**Main Review:**

The Gibbs algorithm is an idealized model of learning and can be thought of as a soft version of the ERM algorithm. The generalization error of the Gibbs algorithm is a well-studied topic and I believe the characterization obtained in this paper is novel and useful. The authors show that the characterization can sometimes yield fast rate convergence bounds (depending on the inverse temperature parameter) under certain assumptions. The presentation can be improved in some places but overall the writing is clear. Below are some comments.

Comments:

-- In l.26-29, the authors write, "As a consequence, both methods fail to capture the fact that generalization error depends strongly on the interplay between the hypothesis class, learning algorithm, and the underlying data-generating distribution." There are many works that addressed this before (e.g., the mutual information-based bounds, Xu and Raginsky, 2017 and related work). This should be acknowledged.

-- For the asymptotic analysis (Section 4) of the single-well case, the regularity conditions (Appendix E.2) for the asymptotic normality of the MLE estimate is almost never satisfied for many practical learning models (e.g., deep learning models, which are singular). A brief comment to that order can be included in relation to the validity of the results in such settings. Ditto for the multi-well case, which again relies on related assumptions (such as a full-rank Hessian).

Formatting:

-- In the References section, follow the standard NeurIPS formatting guidelines. ref [12] has a different format than similar ones from arxiv/preprints.

**** Post-rebuttal:

I have read the authors' response and maintain my original score.

**Time Spent Reviewing:**

two and half

---

> ### Author Response · Authors · 2021-08-09
> **Responding to Reviewers Comments**
>
> Thank you for taking the time to thoroughly read and comment on our paper. We agree the exposition can be improved in places, and we will do so in the revised version.
>
> $\textbf{Reviewer's Comment:}$ In l.26-29, the authors write, "As a consequence, both methods fail to capture the fact that generalization error depends strongly on the interplay between the hypothesis class, learning algorithm, and the underlying data-generating distribution." There are many works that addressed this before (e.g., the mutual information-based bounds, Xu and Raginsky, 2017 and related work). This should be acknowledged.
>
> $\textbf{Response:}$ Good point.  We will include reference to the work of Xu and Raginsky (2017) in the revised manuscript, placing it after the sentence mentioned by the reviewer in l.26-29.
>
> $\textbf{Reviewer's Comment:}$ For the asymptotic analysis (Section 4) of the single-well case, the regularity conditions (Appendix E.2) for the asymptotic normality of the MLE estimate is almost never satisfied for many practical learning models (e.g., deep learning models, which are singular). A brief comment to that order can be included in relation to the validity of the results in such settings. Ditto for the multi-well case, which again relies on related assumptions (such as a full-rank Hessian).
>
> $\textbf{Response:}$
> To clarify, our propositions 1 and 2 only require the full-rank Hessian assumption, and the regularity conditions in Appendix E.2 are only used for obtaining (21). It is true that many practical learning models do not have full-rank Hessian matrix. However, the full-rank assumption is adopted due to the technical simplicity of Gaussian approximation and to provide a closed-form result.
>  The full-rank Hessian assumption can be relaxed by using Theorem 6.2 in [11], which characterizes the limiting distribution of the Gibbs algorithm by considering singular Hessian matrix. It is shown in [11] that the limiting distribution with a singular Hessian is non-Gaussian.
>
>  We will include this broader discussion of the validity of the assumptions and possible extensions of Proposition 1 and 2 in the revised paper, which we agree that readers will appreciate.
>
> $\textbf{Reviewer's Comment:}$ Formatting: In the References section, follow the standard NeurIPS formatting guidelines. ref [12] has a different format than similar ones from arxiv/preprints.
>
>  $\textbf{Response:}$ Thanks for catching this. The format of reference [12] will be corrected in the revised manuscript.  And we will check carefully for other formatting errors in preparing the revised manuscript.

---

### Official Review · Reviewer_3ZSG · 2021-07-16

**Rating:** 7
**Confidence:** 2

**Summary:**

This paper shows the generalization error (actually, the population loss minus the empirical  loss) of the Gibbs algorithm is the symmetric KL divergence between the sample distribution and the Gibbs distribution on hypotheses divided by the inverse temperature.  This formulation is then used to tighten existing generalization bounds on the Gibbs algorithm in several settings.  It allows for improved bounds as the inverse temperature $\alpha$ goes to infinity (and the Gibbs algorithm approaches empirical risk minimization), as existing bounds on the Gibbs algorithm seem to behave as $O( \alpha / n)$.   The author's also consider regularized Gibbs algorithms and use their formulation to deduce when the regularizer will be beneficial.


**Limitations And Societal Impact:**

It seems unlikely that the techniques would generalize beyond the Gibbs algorithm, I didn't understand the concluding comment about over-parameterized Gibbs algorithms.



**Main Review:**

The paper appears to present an advance in the analysis of the Gibbs algorithm, however I am not conversant enough with the previous work to be confident of its originality.  The improvement of other bounds seems interesting and significant.

The quality of the work seems good - with only light checking it seems to be technically correct with appropriate techniques.
The focus on the important Gibbs algorithm is well-motivated, and seems to connect with a variety of previous work, but the
results do seem restricted to that algorithm.

 It is generally clear and well-written, but is a little heavy.  That seems to be the case with most information theoretical work.
When reading the improved $\alpha \rightarrow \infty$ bounds, I wondered how they compared with bounds on ERM in those settings.

**Time Spent Reviewing:**

~3 hours

---

> ### Author Response · Authors · 2021-08-09
> **Responding to Reviewers Comments**
>
> Thank you for taking the time to thoroughly read and comment on our paper.
>
> $\textbf{Reviewer's Comment:}$ I am not conversant enough with the previous work to be confident of its originality.
>
> $\textbf{Response:}$ One of our main contributions is the $\textit{exact}$ characterization of expected generalization error of the Gibbs algorithm in terms of Symmetrized KL information ("Theorem 1") in comparison to all previous works which provide only upper bounds on generalization error or excess risk.
>
> $\textbf{Reviewer's Comment:}$ When reading the improved $\alpha \rightarrow \infty$ bounds, I wondered how they compared with bounds on ERM in those settings.
>
> $\textbf{Response:}$ In the regime $\alpha \rightarrow \infty$, the Gibbs algorithm will converge to ERM algorithm, and Proposition 1 provides an exact characterization (not a bound) of the generalization error in the single-well case. In order to obtain a closed form expression, we construct an  upper bound of generalization error for ERM in the multiple-well case in Proposition 2, which utilizes the data-processing inequality to simplify the computation of symmetrized KL information. Thus, our bound should be tighter compared to standard bounds obtained based on uniform convergence (see [64]). We will clarify these points in the revised manuscript.
>
> $\textbf{Reviewer's Comment:}$ I didn't understand the concluding comment about over-parameterized Gibbs algorithms.
>
> $\textbf{Response:}$ In the future, we want to extend these results (Proposition 1 and Proposition 2)  to the over-parameterized regime, where both the dimension of the weights $d$ and the number of samples $n$ go to infinity and the Hessian matrix can be singular. We should have been clearer about this in our comment, and will do so in the revised paper.

---

### Official Review · Reviewer_pGbj · 2021-07-22

**Rating:** 7
**Confidence:** 3

**Summary:**

This paper obtains an exact characterization of the expected generalization error of the celebrated Gibbs distribution (algorithm) using symmetrized KL information between the input training samples and the output hypothesis. The paper then obtains tighter generalization error bounds for the Gibbs distribution and studies the asymptotic behavior of that distribution when the temperature tends to zero. The effect of regularization on the generalization error of the Gibbs distribution is also studied.

**Limitations And Societal Impact:**

The authors have adequately addressed the limitations and potential negative societal impact of their work.

**Main Review:**

The paper has a nice contribution in writing the expected generalization error of the Gibbs distribution as an information measure, i.e., the symmetrized KL information. It contributes to the recent line of work on the intersection of machine learning and information theory, more specifically information-theoretic generalization bounds. The proofs look correct to me, however, I did not verify them completely. The paper is well-written and well-organized, and compares the results to prior work in the literature. My only complaint with respect to citing prior works is that the paper seems to have missed in lines 128-133 mentioning the results and excess risk bounds on the Gibbs distribution found in [10, Appendix D]. The well-known paper by Jaynes “Information theory and statistical mechanics” (1957) which proves that the Gibbs distribution is the maximizer of energy minus entropy is better to be cited as well.

Minor comments:
1. In line 160, it is more convenient for the reader to mention exactly which result of [36] deals with non-negativity.
2. The proof of Theorem 4 is simple and incremental with respect to Theorem 1. Thus, it is better to be stated either as a corollary or a proposition.


**Time Spent Reviewing:**

5

---

> ### Author Response · Authors · 2021-08-09
> **Responding to Reviewers Comments**
>
> Thank you for taking the time to thoroughly read and comment on our paper.
>
> $\textbf{Reviewer's Comment:}$ My only complaint with respect to citing prior works is that the paper seems to have missed in lines 128-133 mentioning the results and excess risk bounds on the Gibbs distribution found in [10, Appendix D].
>
> $\textbf{Response:}$ Thanks, we will add discussion of the excess risk upper bounds on the Gibbs algorithm in [10, Appendix D] and related works in the revised manuscript.
>
> $\textbf{Reviewer's Comment:}$ The well-known paper by Jaynes “Information theory and statistical mechanics” (1957) which proves that the Gibbs distribution is the maximizer of energy minus entropy is better to be cited as well.
>
> $\textbf{Response:}$ Thanks, we will include a reference to “Information theory and statistical mechanics” (1957) by Jaynes in the revised manuscript.
>
> $\textbf{Reviewer's Comment:}$ In line 160, it is more convenient for the reader to mention exactly which result of [36] deals with non-negativity.
>
> $\textbf{Response:}$ The non-negativity of the generalization error is not mentioned explicitly in [36]. However, as shown in the proof of Theorem 1 in [36], (see Appendix A.2 equation (35)), the expected generalization error can be lower bounded by a KL divergence term under sub-Gaussian assumption, which is always non-negative. We will make this point clear in the revised manuscript.
>
> $\textbf{Reviewer's Comment:}$ The proof of Theorem 4 is simple and incremental with respect to Theorem 1. Thus, it is better to be stated either as a corollary or a proposition.
>
> $\textbf{Response:}$ Good point. We will change "Theorem 4" to "Proposition 3" in the revised manuscript.

---

### Decision · Program_Chairs · 2021-09-27

**Decision:**

Accept (Poster)

**Comment:**

This paper studies the Gibbs posterior (called Gibbs algorithm in the paper), an extension of the posterior in Bayesian statistics, where the negative log-likelihood is replaced by a general loss function. The Gibbs posterior is motivated by information theory, PAC-Bayes bounds and many approaches that aim at generalizing the Bayesian techniques in ML.

The main result of the paper (Theorem 1) is an exact characterization of the Gibbs posterior expected risk, as the symmetrized KL information between the sample and the parameter over the inverse-temperature.

The authors then provide applications of this result: improvements of existing PAC-Bayes bounds in the sub-Gaussian case (Section 4), and asymptotic study in this case where the inverse temperature grows to infinity (Section 5; this case is not covered by existing PAC-Bayes bounds).

Finally, in Section 6, the author(s) generalize Theorem 1 to the case where one adds a possibly data-dependent regularizer to the loss function.

Theorem 1 is new. It leads to interesting generalization or improvements of existing results. The paper should be of interest for researchers not only in the PAC-Bayes community, but more generally for all researchers in Bayesian ML. My opinion is that the paper is overal very well written. This is also the opinion of all the Reviewers, even though some of them mentioned that the paper is a little dense in parts. Some of them raised minor points / typos, please take them into account.

I would just point out what I believe is a major terminology problem. The probability distribution in (3) is sometimes refered to as "pseudo-posterior", "generalized posterior", "Gibbs posterior" (e.g. [4]) or "Gibbs estimator" (e.g. [19]). It is sipply a Gibbs distribution, but "posterior" or "estimator" emphasize the fact that in this case, this distribution is used as a tools for statistics / machine learning. This is the first case I see it refered to as "Gibbs algorithm", and this is problematic for two reasons:
- this is not an algorithm,
- the term "Gibbs algorithm" is sometimes used to refer to the "Gibbs sampling" algorithm https://en.wikipedia.org/wiki/Gibbs_sampling introduced and studied by [Geman, S., & Geman, D. (1984). Stochastic relaxation, Gibbs distributions, and the Bayesian restoration of images. IEEE Transactions on pattern analysis and machine intelligence, (6), 721-741].
To keep calling "Gibbs algorithm" the generalized posterior in (3) would thus lead to confusion. I strongly recommend to the authors to change for "generalized posterior", "Gibbs posterior" or even "Gibbs distribution".

For these reasons, I strongly recommend to change the name of the paper from "Gibbs algorithm", for example to "Gibbs posterior".